# A Three-Dimensional Integrated Non-Linear Coordinate Control Framework for Combined Yaw- and Roll-Stability Control during Tyre Blow-Out

**DOI:** 10.3390/s21248328

**Published:** 2021-12-13

**Authors:** Boyuan Li, Chao Huang, Yang Wu, Bangji Zhang, Haiping Du

**Affiliations:** 1State Key Laboratory of Advanced Design and Manufacturing for Vehicle Body, Hunan University, Changsha 410082, China; bl995@uowmail.edu.au (B.L.); kisera@126.com (Y.W.); bangjizhang@hnu.edu.cn (B.Z.); 2Department of Industrial and Systems Engineering, The Hong Kong Polytechnic University, Hong Kong; 3School of Electrical, Computer and Telecommunications Engineering, University of Wollongong, Wollongong, NSW 2522, Australia; hdu@uow.edu.au

**Keywords:** tyre blow-out, yaw stability, roll stability, vehicle dynamics model, model predictive control

## Abstract

A tyre blow-out can greatly affect vehicle stability and cause serious accidents. In the literature, however, studies on comprehensive three-dimensional vehicle dynamics modelling and stability control strategies in the event of a sudden tyre blow-out are seriously lacking. In this study, a comprehensive 14 degrees-of-freedom (DOF) vehicle dynamics model is first proposed to describe the vehicle yaw-plane and roll-plane dynamics performance after a tyre blow-out. Then, based on the proposed 14 DOF dynamics model, an integrated control framework for a combined yaw plane and roll-plane stability control is presented. This integrated control framework consists of a vehicle state predictor, an upper-level control mode supervisor and a lower-level 14 DOF model predictive controller (MPC). The state predictor is designed to predict the vehicle’s future states, and the upper-level control mode supervisor can use these future states to determine a suitable control mode. After that, based on the selected control mode, the lower-level MPC can control the individual driving actuator to achieve the combined yaw plane and roll plane control. Finally, a series of simulation tests are conducted to verify the effectiveness of the proposed control strategy.

## 1. Introduction

A sudden vehicle tyre blow-out may cause significant problems to vehicle stability and road safety. In the United States (US), the published statistical data shows ‘tyre blow-out’ caused more than 300,000 road accidents in the years 1992 to 1996 [1]. Based on the data from the report by the National Highway Traffic Safety Administration (NHTSA) in the US, tyre blow-outs caused 414 fatalities, 10,275 nonfatal injuries, and 78,392 crashes in 2003 [2]. In addition, tyre blow-outs also cause serious stability issues in electric industrial vehicles, such as forklift trucks [3,4].

The blow-out of one specific tyre makes the tyre pressure significantly decrease and causes a significant change to the vehicle’s dynamic response. Various studies have proved that a tyre blow-out can be completed within 0.1 s and the tyre parameter change can be considered as a step change [5,6]. It is argued that the tyre deflation greatly affects cornering stiffness, radial tyre stiffness and rolling resistance [5,7]. In [8], actual experiments on 26 vehicles were carried out to study the vehicles’ dynamic response to tyre blow-outs. The experiment results suggested that the increased rolling resistance of deflated tyres could generate longitudinal drag force and cause additional yaw moment to pull the vehicle away from the original path. The studies [6,9] also pointed out that the tyre cornering stiffness and radial stiffness decreased significantly after tyre blow-out. The assumption of the tenfold drop of radial stiffness after tyre blow-out was verified by the tests on a 165SR13 D90 tyre, and the decreased tyre radial stiffness caused the tyre’s instantaneous radius reduction and significantly increased the load transfer effect. It is suggested in [5] that tyre cornering stiffness and radial stiffness reduces by 25–40% after tyre deflation. Similarly, Wang et al. proposed a non-linear coordinate motion controller for the vehicle after tyre deflation by assuming the rolling resistance increased 30 times and the cornering stiffness reduced to 28% of the original value [9]. In addition, when a tyre blow-out happens, at the steering wheel more steering input is required to compensate for the increased total alignment moment caused by the deflated tyre [8], and the steering controller needs to be redesigned, for instance, with the human-machine adaptive shared control [10]. However, the steering control system design is not focused on in this study. Based on the review of the above studies, it can be summarised that the tyre blow-out mainly affects vehicle dynamics performance in three aspects: (1) the additional yaw moment is induced by the increased rolling resistance of the deflated tyre; (2) the changed tyre lateral force is caused by the decreased tyre cornering stiffness; (3) the decreased radial stiffness will cause a significant decrease of the wheel’s instantaneous radius and induce a big load transfer effect.

In current literature, a number of studies have proposed different kinds of vehicle dynamics models to present the dynamics performance after tyre blow-out. In [6,7], the three-dimensional Engineering Dynamics Vehicle Simulation Model (EDVSM) is used to describe tyre blow-out behaviour. This comprehensive vehicle model has 15 degrees of freedom (DOF): 6 DOF for the vehicle body, 4 DOF for the suspension system, 4 DOF for the wheel rotation and one DOF for the steering wheel. Similarly, the high-order comprehensive commercial vehicle dynamics model veDYNA is applied in [5,11] to present the vehicle dynamics performance after tyre blow-out. However, the EDVSM and veDYNA vehicle models are all commercial products and the detailed mathematical equations of these models are not presented, so it is hard to carry out the theoretical study on the tyre blow-out modelling. The stability controller design after a tyre blow-out in studies [5,9,11] is only based on the yaw plane dynamics equation (only considering the changed rolling resistance and cornering stiffness after tyre blow-out), and the suspension motion and vertical dynamics have been neglected. When one specific tyre blows out, the suddenly decreased tyre radial stiffness will cause the reduction of the instantaneous tyre radius. This reduction will transfer to the suspension system and cause a big suspension deflection, load transfer and increase of the roll angle. This will cause a strong coupling effect on the yaw plane dynamics and should be considered in the controller design. Therefore, a three-dimensional full-vehicle dynamics model, which considers all six degrees of freedom of the vehicle body (longitudinal motion, lateral motion, vertical motion, yaw motion, roll motion and pitch motion) and integrating the suspension system and vehicle body dynamics system, is required to comprehensively present the dynamics response of a vehicle after one specific tyre blow-out for the stability controller design.

In the current vehicle industry, the tyre pressure monitoring system (TPMS) based on new in-tyre sensors and electronics is widely used to monitor the tyre pressure in real-time and detect tyre blow-out early [2,12]. Although some studies have proposed fault diagnosis and estimation approaches in the literature [13], we can simply assume the location of tyre blow-out is already known. After the blow-out of a specific tyre has been detected, various vehicle stability control systems are designed to improve the vehicle handling and stability. The control algorithms in the literature can be classified into three types: the steering-only control, the braking-only control and the integrated control. In [14], a steering-only control approach is presented, and the control system is triggered by an alarm generated by the TPMS. Chen et al. proposes the control strategy for an emergency automatic braking system when a tyre blows out [15]. Wang et al. developed a control optimization strategy for the yaw-plane motion by coordinating both the steering and braking based on a triple-step control method: steady-state controller, feedforward controller and feedback controller [9,11]. In [9,11], the longitudinal vehicle dynamics are neglected and the longitudinal velocity is assumed to be available from the estimation algorithm. However, the time-varying longitudinal velocity will greatly affect the vehicle handling and stability after a tyre blow-out and the effect of the changing longitudinal velocity on the controller design should not be neglected. In [5], the gain scheduling robust controller with respect to time-varying longitudinal velocity after tyre blow-out is proposed. The feedback control gain of a high-level controller can be real-time adjusted by the changing scaling factors determined by different values of current longitudinal velocity value and maximum and minimum velocity values.

The above studies [5,9,10,11] focus on the yaw-plane stability control during tyre blow-out and the main control targets are the side-slip angle and yaw rate. However, a tyre blow-out strongly affects the vehicle roll dynamics and the roll plane control targets should be also included in the controller design. Currently, a rollover can be mitigated by using the brakes [16,17,18], steering [19,20], antiroll bars [21] or a combination of different actuators [22,23]. Some of the current studies have discussed the combined control of yaw stability and roll stability. For example, Rajamani et al. carried on a study to explore the vehicle yaw and roll dynamics response in the steady-state turning manoeuvre [24]. It is concluded that in steady-state cornering, the roll angle and rollover index remain unchanged unless the longitudinal velocity or the cornering trajectory is changed. Alberding et al. propose a non-linear hierarchical control allocation algorithm for vehicle yaw stabilisation and rollover prevention by using differential braking, and this controller eliminates the roll controller by introducing the rollover prevention as a constraint in the control algorithm [25].

Model predictive control (MPC) can predict the vehicle’s future state and is greatly advantageous in rollover prevention. In addition, MPC is suitable for dealing with multiple control targets within defined constraints. Yin et al. propose a non-linear MPC to achieve the path-tracking control by utilising the prediction horizon of MPC [26]. Similarly, Chen et al. also design an LQR lateral control method based on the optimal front tyre lateral force [27].

A recent study proposes a combined yaw and roll-stability control framework based on the MPC method [28]. In [28], however, only the control actuator of differential braking is utilised to achieve various control targets, which limits the control performance. In [29], an MPC control system is proposed by integrating lateral stability control, rollover prevention and longitudinal slip control. Furthermore, an integrated control system based on fuzzy differential braking is developed to improve the yaw and rollover stability of off-road vehicles [30]. The new emerging technology of electric vehicles with in-wheel motors can achieve four-wheel-independent-driving (4WID) and the driving or braking torque can be optimally controlled and allocated to the individual wheel and the control envelope is substantially enlarged. A number of studies have proposed utilising the 4WID function to achieve better dynamic stability control performance [31,32].

In this study, first a 14 DOF vehicle dynamics model including the yaw-plane motion, roll-plane motion, pitch-plane motion and suspension dynamics is proposed, which is utilised to present the impact of abruptly changed tyre rolling resistance, cornering stiffness and vertical stiffness after a tyre blow-out on vehicle dynamics performance. Then, based on the comprehensive dynamics model, a three-dimensional MPC control allocation framework for integrated yaw-plane stability and roll-stability control after tyre blow-out is proposed. Based on a 4WID electric vehicle, this control framework can optimally distribute the driving and braking torque of individual wheels and achieve cruise control, yaw-plane stability control and roll-stability control simultaneously. The proposed control framework has a two-layer control structure and has three control modes: cruise control mode, yaw stability control mode and roll-stability control mode. In the upper-level control strategy, a model predictor is proposed to predict the vehicle’s future states, and a control mode supervisor can determine the suitable control mode based on the predicted states. In the lower level, a MPC controller is applied to allocate the control actuators based on the selected control mode.

The major contribution of our study can be summarised as follows:

(1) A comprehensive 14 DOF dynamic model is applied to describe the vehicle dynamics performance during tyre blow-out, which is less focused on in the literature.

(2) A new integrated yaw- and roll-stability MPC controller based on the 14 DOF model is proposed specifically for the tyre blow-out scenario.

The rest of this paper is organised as follows. First, a 14 DOF vehicle dynamics model is presented in Section 2 to describe the dynamics performance of a tyre blow-out. Then, in Section 3, the simulation results of the dynamics performance of the 14 DOF model and the 8 DOF model after tyre blow-out are compared with the EDVSM model which has been validated by actual experimental results. Section 4 describes the proposed integrated yaw-stability and roll-stability control framework based on MPC. Finally, the simulation results of vehicle control performance during tyre blow-out are presented to validate the proposed control framework.

## 2. Vehicle Dynamics Model Considering the Tyre Blow-Out Effect

### 2.1. Vehicle Body Dynamics Model

In this section, the comprehensive 14 DOF vehicle dynamics model is proposed to present the actual vehicle dynamics performance after tyre blow-out [33] and the detailed diagram description is shown in Figure 1.

The equations of motion of the vehicle sprung mass can be presented as following the six DOF model:(1a)$$m_{s}\left({\dot{v}}_{x}-\omega_{y}v_{z}-\omega_{z}v_{y}\right)=\sum \left(F_{xsi}\right)+m_{s}g\sin\theta$$
(1b)$$m_{s}\left({\dot{v}}_{y}+\omega_{z}v_{x}-\omega_{x}v_{z}\right)=\sum \left(F_{ysi}\right)-m_{s}g\sin\phi \cos\theta$$
(1c)$$m_{s}\left({\dot{v}}_{z}+\omega_{x}v_{y}-\omega_{y}v_{x}\right)=\sum F_{zsij}-m_{s}g\cos\phi \cos\theta$$
(1d)$$J_{x}{\dot{\omega }}_{x}+\left(J_{z}-J_{y}\right)\omega_{y}\omega_{z}=\sum \left(M_{xi}\right)+m_{s}gH_{roll}\sin\phi +\frac{c\left(F_{zs1}-F_{zs2}+F_{zs3}-F_{zs4}\right)}{2}$$
(1e)$$J_{y}{\dot{\omega }}_{y}+\left(J_{x}-J_{z}\right)\omega_{z}\omega_{x}=\sum \left(M_{yi}\right)+l_{r}\left(F_{zs3}+F_{zs4}\right)-l_{f}\left(F_{zs1}+F_{zs2}\right)$$
(1f)$$J_{z}{\dot{\omega }}_{z}+\left(J_{y}-J_{x}\right)\omega_{x}\omega_{y}=l_{f}\left(F_{ys1}+F_{ys2}\right)-l_{r}\left(F_{ys3}+F_{ys4}\right)+\frac{c\left(-F_{xs1}+F_{xs2}-F_{xs3}+F_{xs4}\right)}{2}$$
where $m_{s}$ is the vehicle sprung mass and g is the acceleration gravity. $J_{x}$, $J_{y}$, $J_{z}$ are inertial moments of pitch, roll and yaw, respectively. $v_{x}$,$\;v_{y}$,$\;v_{z}$ are longitudinal velocity, lateral velocity and vertical velocity, respectively. $\omega_{x}$,$\;\omega_{y}$,$\;\omega_{z}$ are pitch rate, roll rate and yaw rate, respectively. $F_{xsi}$,$\;F_{ysi}$, $F_{zsi}$ represent the longitudinal force, lateral force and vertical force transferred to C.G. in the coordinate system attached to C.G. i=1,2,3,4, which presents the front left, front right, rear left and rear right wheel. $F_{dzij}$ shows the load transfer force of each wheel. $l_{f}$ is the front wheelbase and $l_{r}$ is the rear wheelbase. c is the track width. $h_{f}$ and $h_{r}$ represent the distance between front and rear roll centres and C.G. $M_{xi}$ and $M_{yi}$ are roll moment and pitch moment transmitted to the sprung mass. $H_{roll}$ is the distance between C.G. and vehicle roll centre of the sprung mass.

The roll angle ϕ, pitch angle θ and yaw angle ψ can be determined as the following equations:(2a)$$\dot{\theta }=\omega_{y}\cos\phi -\omega_{z}\sin\phi$$
(2b)$$\dot{\psi }=\frac{\omega_{y}\sin\phi }{\cos\theta }+\frac{\omega_{z}\cos\phi }{\cos\theta }$$
(2c)$$\dot{\phi }=\omega_{x}+\omega_{y}\sin\phi \tan\theta +\omega_{z}\cos\phi \tan\theta$$

The tyre force $F_{xsi}$ and $F_{ysi}$ can be determined by subtracting the unsprung mass weight and inertial force from the corresponding forces acting on the tyre contact patch:(3a)$$F_{xsi}=F_{xgsi}+m_{ui}g\sin\theta -m_{ui}{\dot{v}}_{xui}+m_{ui}\omega_{z}v_{yui}-m_{ui}\omega_{y}v_{zui}$$
(3b)$$F_{ysi}=F_{ygsi}-m_{ui}g\sin\phi \cos\theta -m_{ui}{\dot{v}}_{yui}+m_{ui}\omega_{x}v_{zui}-m_{ui}\omega_{z}v_{xui}$$
where $m_{ui}$ is the unsprung mass of an individual corner. $v_{xui}$,$\;v_{yui}$,$\;v_{zui}$ are unsprung mass longitudinal velocity/lateral velocity/vertical velocity in a coordinate system attached to C.G. $F_{xgsi}$, $F_{ygsi}$, $F_{zgsi}$ are tyre–road contact forces in the body-fixed coordinate system, which can be projected from the $F_{xgi}$, $F_{ygi}$, $F_{zgi}$ (tyre force in the coordinate system fixed at the tyre contact patch) as:(4)$$\left[\begin{array}{c} F_{xgsi}\\ F_{ygsi}\\ F_{zgsi} \end{array}\right]=\left[\begin{array}{ccc} 1 & 0 & 0\\ 0 & \cos\phi & \sin\phi \\ 0 & -\sin\phi & \cos\phi \end{array}\right]\left[\begin{array}{ccc} \cos\theta & 0 & -\sin\theta \\ 0 & 1 & 0\\ \sin\theta & 0 & \cos\theta \end{array}\right]\left[\begin{array}{c} F_{xgi}\\ F_{ygi}\\ F_{zgi} \end{array}\right]$$

The roll moment $M_{xi}$ and pitch moment $M_{yi}$ can be determined by the following equations:(5a)$$M_{x1}=F_{ys1}H_{roll}$$
(5b)$$M_{x2}=F_{ys2}H_{roll}$$
(5c)$$M_{x3}=F_{ys3}H_{roll}$$
(5d)$$M_{x4}=F_{ys4}H_{roll}$$
(5e)$$M_{yi}=-\left(F_{xsgi}R_{i}+F_{xsi}l_{si}\right)$$
where $R_{i}$ is the instantaneous length of tyre radius and $l_{si}$ is the instantaneous length of strut.

The vertical tyre force $F_{zsi}$ can be determined according to the following equation:(6a)$$F_{zs1}=k_{sf}x_{s1}+b_{sf}{\dot{x}}_{s1}-\frac{M_{ARB\_F}}{c}$$
(6b)$$F_{zs2}=k_{sf}x_{s2}+b_{sf}{\dot{x}}_{s2}+\frac{M_{ARB\_F}}{c}$$
(6c)$$F_{zs3}=k_{sr}x_{s3}+b_{sr}{\dot{x}}_{s3}-\frac{M_{ARB\_R}}{c}$$
(6d)$$F_{zs4}=k_{sr}x_{s4}+b_{sr}{\dot{x}}_{s4}+\frac{M_{ARB\_R}}{c}$$
where $x_{si}$ is the suspension spring compression. $k_{sf},k_{sr}$ are suspension stiffness and $b_{sf}$,$\;b_{sr}$ are suspension damping coefficient. The anti-roll moment from the anti-roll bar can be determined by:(7a)$$M_{ARB\_F}=0.5k_{ARB,f}\left(x_{s1}-x_{s2}\right)+0.5b_{ARB,f}\left({\dot{x}}_{s1}-{\dot{x}}_{s2}\right)$$
(7b)$$M_{ARB\_R}=0.5k_{ARB,r}\left(x_{s3}-x_{s4}\right)+0.5b_{ARB,r}\left({\dot{x}}_{s3}-{\dot{x}}_{s4}\right)$$
where $k_{ARB,f}$,$\;k_{ARB,r}$ are the stiffness of anti-roll bar and $b_{ARB,f}$,$\;b_{ARB,r}$ are the damping coefficient of the anti-roll bar.

The jacking force $F_{dzi}$ transmitted to the sprung mass through the struts can be calculated as:(8a)$$F_{dz2}=-F_{dz1}=\frac{F_{ygs1}R_{1}+F_{ygs2}R_{2}+F_{ys1}l_{s1}+F_{ys2}l_{s2}-\left(F_{ys1}+F_{ys2}\right)H_{roll}}{c}$$
(8b)$$F_{dz4}=-F_{dz3}=\frac{F_{ygs3}R_{3}+F_{ygs4}R_{4}+F_{ys3}l_{s3}+F_{ys4}l_{s4}-\left(F_{ys3}+F_{ys4}\right)H_{roll}}{c}$$

### 2.2. Tyre Model

The non-linear Dugoff tyre model is used in this paper to present the tyre’s non-linear characteristics and determine the tyre longitudinal force $F_{xti}$ and lateral force $F_{yti}\;$[34,35], and is described by:(9a)$$\lambda_{i}=\frac{\mu F_{zgi}\left[1-\varepsilon_{r}v_{si}\sqrt{s_{i}^{2}+\tan^{2}\alpha_{i}}\right]\left(1-s_{i}\right)}{2\sqrt{C_{s}^{2}s_{i}^{2}+C_{\alpha i}^{2}\tan^{2}\alpha_{i}}}$$
(9b)$$f\left(\lambda_{i}\right)=\{\begin{array}{c} \lambda_{i}\left(2-\lambda_{i}\right)\;\left(\lambda_{i}<1\right)\;\\ 1\;\left(\lambda_{i}>1\right) \end{array}$$
(9c)$$F_{yti}=\frac{C_{\alpha }\tan\alpha_{i}}{1-s_{i}}f\left(\lambda_{i}\right)$$
(9d)$$F_{xti}=\frac{C_{s}s_{i}}{1-s_{i}}f\left(\lambda_{i}\right)$$
where μ is the tyre–road friction coefficient.$\;C_{s}$ is the longitudinal cornering stiffness and $C_{\alpha i}$ is the lateral cornering stiffness of each wheel. $\varepsilon_{r}$ is a constant value. The side-slip angle $\alpha_{i}$ and slip ratio $s_{i}$ of the individual tyres can be calculated as the following:(10a)$$\alpha_{1}=\delta_{1}-\tan^{-1}\left(\frac{v_{yg1}}{v_{xg1}}\right)$$
(10b)$$\alpha_{2}=\delta_{2}-\tan^{-1}\left(\frac{v_{yg2}}{v_{xg2}}\right)$$
(10c)$$\alpha_{3}=\delta_{3}-\tan^{-1}\left(\frac{v_{yg3}}{v_{xg3}}\right)$$
(10d)$$\alpha_{4}=\delta_{4}-\tan^{-1}\left(\frac{v_{yg4}}{v_{xg4}}\right)$$
(11a)$$v_{sfl}=\cos\delta_{fl}\left(v_{xg1}\right)+\sin\delta_{fl}\left(v_{yg1}\right)$$
(11b)$$v_{sfr}=\cos\delta_{fr}\left(v_{xg2}\right)+\sin\delta_{fr}\left(v_{yg2}\right)$$
(11c)$$v_{srl}=\cos\delta_{rl}\left(v_{xg3}\right)+\sin\delta_{rl}\left(v_{yg3}\right)$$
(11d)$$v_{srr}=\cos\delta_{rr}\left(v_{xg4}\right)+\sin\delta_{rr}\left(v_{yg4}\right)$$
(12)$$s_{i}=\frac{\omega_{i}R_{i}-v_{si}}{max\left(\omega_{i}R_{i},v_{si}\right)}$$

Longitudinal velocity and lateral velocity at tyre contact patch $v_{xgi}$ and $v_{ygi}$ can be presented as the following equations:(13a)$$v_{xgi}=\cos\theta \left(v_{xui}-\omega_{y}R_{i}\right)+\sin\theta \left(v_{zui}\cos\phi +\sin\phi \left(\omega_{x}R_{i}+v_{yui}\right)\right)$$
(13b)$$v_{ygi}=\cos\phi \left(v_{yui}+\omega_{x}R_{i}\right)-v_{zui}\sin\phi$$

$F_{zgi}$ is the vertical force acting on the tyre–ground contact patch, which can be calculated by the following equation:(14)$$F_{zgi}=k_{ti}x_{ti}$$

$k_{ti}$ is the tyre vertical stiffness. $x_{ti}$ is the tyre spring compression and the initial tyre compression:(15)$$x_{t0}=\frac{\frac{ml_{r}}{2\left(l_{f}+l_{r}\right)}+m_{ui}}{k_{t}}$$

The velocity of the tyre’s instantaneous deflection can be calculated as the following:(16)$${\dot{x}}_{ti}=v_{xui}\sin\theta -\cos\theta \left(v_{zui}\cos\phi +v_{yui}\sin\phi \right)$$

The instantaneous tyre radius can be calculated as:(17)$$R_{i}=\frac{R_{0}-x_{ti}}{\cos\theta \cos\phi }$$
where $R_{0}$ is the nominal tyre radius.

The wheel dynamics equations can be presented as follows:(18)$$I_{\omega }{\dot{\omega }}_{i}=-R_{i}F_{xti}+T_{i}-M_{yi}$$
where $I_{\omega }$ is the wheel rotational inertia. $\omega_{i}$ is the wheel angular speed and $T_{i}$ is the traction/brake torque of the individual wheel. $M_{yi}$ is the rolling resistance moment, which can be presented by the following equation:(19)$$M_{yi}=F_{zgi}\left(K_{fi}+K_{fvi}v_{xgi}{}^{2}\right)R_{i}$$
where $K_{fi},K_{fvi}$ are the tyre rolling resistance coefficients of the individual tyre.

### 2.3. Suspension System

The instantaneous compression of the suspension spring $x_{si}$ can be calculated by the following equation:(20)$${\dot{x}}_{si}=-v_{zsi}+v_{zui}$$
where $v_{zsi}$ is the vertical velocity of the strut mounting point of each wheel, which can be calculated as the following equation:(21a)$$v_{zs1}=v_{z}+\frac{c\omega_{x}}{2}-l_{f}\omega_{y}$$
(21b)$$v_{zs2}=v_{z}-\frac{c\omega_{x}}{2}-l_{f}\omega_{y}$$
(21c)$$v_{zs3}=v_{z}-\frac{c\omega_{x}}{2}+l_{r}\omega_{y}$$
(21d)$$v_{zs4}=v_{z}+\frac{c\omega_{x}}{2}+l_{r}\omega_{y}$$

The unsprung mass vertical velocity $v_{zui}$ can be calculated as:(22)$$m_{u}{\dot{v}}_{zui}=\cos\phi \left(\cos\theta \left(F_{zgi}-m_{ui}g\right)+\sin\theta F_{xgi}\right)-\sin\phi F_{ygi}-F_{dzi}-x_{si}k_{si}-{\dot{x}}_{si}b_{si}-m_{ui}\left(v_{zui}\omega_{x}-v_{xui}\omega_{y}\right)$$

Forces $F_{xgi}$ and $F_{ygi}$ can be obtained by the following equation:(23a)$$F_{xgi}=F_{xti}\cos\delta_{i}-F_{yti}\sin\delta_{i}$$
(23b)$$F_{ygi}=F_{yti}\cos\delta_{i}+F_{xti}\sin\delta_{i}$$

The longitudinal and lateral velocities $v_{xui}$, $v_{yui}$ of unsprung mass can be calculated as:(24a)$$v_{xui}=v_{xsi}-l_{si}\omega_{y}$$
(24b)$$v_{yui}=v_{ysi}+l_{si}\omega_{x}$$
where longitudinal and lateral velocity of the strut mounting point of each wheel ($v_{xsi}$ and $v_{ysi}$) can be calculated as the following equation:(25a)$$v_{xs2}=v_{xs4}=v_{x}+\frac{c}{2}\omega_{z}$$
(25b)$$v_{xs1}=v_{xs3}=v_{x}-\frac{c}{2}\omega_{z}$$
(25c)$$v_{ys1}=v_{ys2}=v_{y}+l_{f}\omega_{z}$$
(25d)$$v_{ys3}=v_{ys4}=v_{y}-l_{r}\omega_{z}$$

The instantaneous length of the strut $l_{si}$ can be calculated as:(26)$$l_{si}=l_{s0}-\left(x_{si}-x_{s0}\right)$$
where the initial strut length $l_{s0}=h-\left(R_{0}-x_{t0}\right)$ and the initial suspension deflection
(27)$$x_{s0}=\frac{ml_{r}}{2\left(l_{f}+l_{r}\right)k_{s}}$$

### 2.4. The Effectiveness of Tyre Blow-Out

The tyre blow-out will cause the sudden increase of the rolling resistance of the deflated tyre and induce an additional yaw moment $T_{b}$. Based on [36], the yaw moment $T_{b}$ caused by a tyre blow-out can be determined by the following equation:(28a)$$T_{b}=0.5c\left(F_{cfl}-F_{cfr}\right)$$
where c is the tracking width of the vehicle, the location of the blow-out tyre is at the front axle or
(28b)$$T_{b}=0.5c\left(F_{crl}-F_{crr}\right)$$
where the location of the blow-out tyre is at the rear axle. $F_{ci}$ presents the tyre rolling resistance force, which can be calculated as the following equation:(29)$$F_{ci}=K_{fi}F_{zgi}$$

It is suggested that the typical rolling resistance stiffness for the light vehicle is around 0.012 and 0.015 [36] and it is argued in [7] that this value increases thirty times after a tyre blow-out. Thus, in this study, the rolling resistance coefficient when the tyre is in a healthy condition is chosen as 0.014 and this value increases to 0.42 after tyre blow-out.

In addition, a tyre blow-out causes a sudden decrease of tyre vertical stiffness $k_{ti}$ and a decrease of the tyre’s instantaneous radius $R_{i}$. The following equation shows the effect of the changed tyre’s vertical stiffness on the vehicle vertical tyre load and suspension system:(30)$$F_{zgi}=k_{1}k_{ti}x_{ti}$$
where $k_{1}$ is the ratio of the changed vertical stiffness related to the tyre deflation. When the tyre is in a healthy condition, $k_{1}=1$; when the tyre has a blow-out, $k_{1}=0.28$ [5]. The changed vertical tyre force $F_{zgi}$ caused by the tyre blow-out can induce a significant load transfer effect.

The tyre cornering stiffness also reduces to 28% of the original value [5], which will greatly affect the tyre cornering force:(31)$$B_{y}=k_{2}B_{y}$$
where $k_{2}$ is the ratio of changed cornering stiffness related to the tyre deflation.

## 3. Simulation Performance of the Vehicle after Tyre Blow-Out

In this section, the simulation test is carried out to present the vehicle dynamics performance after tyre blow-out based on the suggested 14 DOF comprehensive vehicle dynamics model. For comparison purposes, the dynamics performance of the widely applied 8 DOF vehicle model which neglects the pitch dynamics motion, vertical dynamics motion and suspension dynamics motion of the four wheels is also presented [34]. Furthermore, the above simulation results are validated against the simulation results from the EDVSM tyre blow-out model proposed in a study [7], where the EDVSM model has been verified by the experimental results of tyre blow-out. The simulation test applies the same straight-line manoeuvre in [7], where the vehicle speed is 101 km/h at the time of a rear right tyre blow-out, and the driver steers and brakes to maintain vehicle control. The tyre–road friction coefficient is assumed as 1. The tyre blow-out happens at 3.8 s and the duration is 0.1 s. The vehicle model parameters applied in the simulation are the same as the values in [7], which is shown in Table 1.

In Figure 2a–d, the longitudinal velocity, yaw rate, longitudinal acceleration and lateral acceleration responses of the 14 DOF model and the 8 DOF model are compared with the results from EDVSM model. After the rear right tyre deflation at 3.8 s, there is a sudden change of the longitudinal velocity, yaw rate, longitudinal and lateral acceleration in both the 8 DOF model and 14 DOF model due to the generated additional yaw moment caused by the sudden increase of the wheel rolling coefficient. It can also be noted that the 14 DOF model shows very similar responses in longitudinal velocity and longitudinal acceleration as the EDVSM model response. The 8 DOF model has a smaller negative longitudinal acceleration response and consequently, the longitudinal velocity is much larger than EDVSM. There are some mismatches of vehicle dynamics responses, such as the yaw rate and lateral acceleration between the 14 DOF model EDVSM model, which required further investigation. The 8 DOF model shows a larger yaw rate and lateral acceleration responses compared with EDVSM model.

Figure 3 also suggests that the instantaneous tyre radius of the deflated rear right tyre of the 14 DOF model at the beginning is smaller than the 8 DOF model due to the tyre compression. The 14 DOF model considers the sudden decrease of tyre vertical stiffness when the tyre blows out, which will induce the significant tyre instantaneous radius reduction. However, the 8 DOF model neglects the tyre vertical dynamics and suspension system and considers the tyre radius as a constant value, which cannot accurately present the tyre blow-out effect.

Figure 4 presents the load transfer effect of 8 DOF model and 14 DOF model after a rear right tyre blow-out. The initial vertical load of 8 DOF model is smaller than 14 DOF model since the 8 DOF model neglects the weight of unsprung mass including the wheel hub, wheel mass and suspension system. For the 8 DOF model, there is no obvious load transfer before and after tyre blow-out happens although the vertical load response of each wheel has a small oscillation during the tyre deflation. On the other hand, the 14 DOF model shows obvious load transfer effect after a front left tyre blow-out: at the beginning of the tyre blow-out, the tyre’s vertical load of the rear right wheel decreases sharply and then a brief spike occurs, which is shown in Figure 4d and is very close to the simulation results from EDVSM model. After that, due to vehicle roll and pitch motion, the tyre’s vertical loads of the front right and rear left wheel are increased and the tyre’s vertical load of the front left is decreased as shown in Figure 4a–c. This vertical load transient response after a rear right tyre blow-out is very similar to the response described in [7].

Figure 5 and Figure 6 compare the yaw-plane stability region of the14 DOF model and the 8 DOF model in different longitudinal velocity conditions. In this study, the yaw-plane stability is determined by the value $\lambda_{i}$ in the Dugoff tyre model: if $\lambda_{i}$ of an individual tyre is larger than 1, the vehicle is in a stable condition; if $\lambda_{i}$ of the individual tyre is equal or smaller than 1, the vehicle is moving in an unstable condition. A group of simulation tests have been carried out to determine the stability transition point when $\lambda_{i}=1$ and consequently the stability boundary can be determined. According to Figure 5 and Figure 6, the stability region of 14 DOF model is generally smaller than 8 DOF model. This is mainly because the 14 DOF model considers the coupling effect of the vehicle roll and pitch motion on the yaw motion and the yaw-plane stability is compromised.

In this section, the proposed 14 DOF model has been validated in the simulation to accurately present the dynamics performance of a tyre blow-out. The value $\lambda_{i}$ in the Dugoff tyre model can be utilised to determine the stability region of the 14 DOF model in the yaw plane. The determined stability region is a very useful tool to select a suitable control mode for the integrated controller design in the following section.

## 4. Three-Dimensional Integrated Yaw-Plane Stability and Roll-Plane Stability 14 DOF MPC Control Framework

In this section, a three-dimensional non-linear coordinate control framework is designed to achieve the integrated control of yaw-plane stability and roll-plane stability when tyre blow-out. Based on 14 DOF model, The hierarchy of the whole 14 DOF MPC control framework consists of the vehicle states predictor, upper-level control supervisor and lower-level 4 DOF MPC controller. Based on predicted vehicle states from a model predictor, the upper-level control mode supervisor selects the most suitable control mode from the options of cruise control mode, yaw-plane stability control mode and roll-stability control mode. Then according to the selected control mode, the lower-level 4 DOF MPC algorithm is applied to allocate the desired control value to the individual actuator. The whole structure of the control framework is shown in Figure 7.

### 4.1. Vehicle States Predictor

A vehicle model predictor based on the model predictive algorithm is presented to determine the vehicle’s future states. Based on some the vehicle’s critical future states, such as the longitudinal velocity, lateral velocity and roll angle, the vehicle control mode can be selected in the upper-level control supervisor.

The major difficulty in implementing the 14 DOF model-based MPC control allocation is the complex model structure of 14 DOF model and the significant increase in computational time. In order to deal with this issue, the 14 DOF model in MPC can be simplified as 4 DOF by assuming some vehicle states are already known or can be directly measured. Therefore, in this section, the vehicle state predictor can be utilised to estimate the vehicle states which cannot be measured directly. Then, the estimated and measured vehicle states can be directly used as input information in the simplified 4 DOF MPC in the lower-level controller.

**Assumption** **1.**
*It is assumed that velocity longitudinal velocity*

$v_{x}$

*, lateral velocity*

$v_{y}$

*and vertical velocity*

$v_{z}$

*in C.G. can be easily estimated [37,38]. Vehicle roll angle*

ϕ

*, roll rate*

$\omega_{x}$

*, pitch angle*

θ

*, pitch rate*

$\omega_{y}$

*, yaw rate*

$\omega_{z}$

*of C.G. and wheel angular velocity of each wheel*

$\omega_{i}$

*are all easy to measure with various sensors. In addition, the tyre cornering stiffness change, tyre vertical stiffness change and rolling resistance change after tyre blow-out are all assumed to be known.*


The model predictive estimator algorithm can be presented in discrete time in this section. The vehicle states which are hard to measure and intended to be estimated are tyre compression $x_{ti}$, suspension spring compression $x_{si}$ and vertical velocity of unsprung mass $v_{zui}$.

First, the velocity of suspension mounting points in the current time step can be calculated based on Equations (21) and (25) in discrete time:(32a)$$v_{xs1}\left(k\right)=v_{xs3}\left(k\right)=v_{x}\left(k\right)-0.5c\omega_{z}\left(k\right)$$
(32b)$$v_{xs2}\left(k\right)=v_{xs4}\left(k\right)=v_{x}\left(k\right)+0.5c\omega_{z}\left(k\right)$$
(33a)$$v_{ys1}\left(k\right)=v_{ys2}\left(k\right)=v_{y}+l_{f}\omega_{z}\left(k\right)$$
(33b)$$v_{ys3}\left(k\right)=v_{ys4}\left(k\right)=v_{y}-l_{r}\omega_{z}\left(k\right)$$
(34a)$$v_{zs1}\left(k\right)=v_{z}\left(k\right)+0.5c\omega_{x}\left(k\right)-l_{f}\omega_{y}\left(k\right)$$
(34b)$$v_{zs2}\left(k\right)=v_{z}\left(k\right)-0.5c\omega_{x}\left(k\right)-l_{f}\omega_{y}\left(k\right)$$
(34c)$$v_{zs3}\left(k\right)=v_{z}\left(k\right)-0.5c\omega_{x}\left(k\right)+l_{r}\omega_{y}\left(k\right)$$
(34d)$$v_{zs4}\left(k\right)=v_{z}\left(k\right)+0.5c\omega_{x}\left(k\right)+l_{r}\omega_{y}\left(k\right)$$

The length of suspension strut in the current time step can be presented based on Equation (26):(35)$$l_{si}\left(k\right)=l_{s0}-\left({\hat{x}}_{si}\left(k\right)-x_{s0}\right)$$

The instance tyre radius in the current time step can be calculated based on Equation (17):(36)$$R_{i}\left(k\right)=\frac{R_{0}-{\hat{x}}_{ti}\left(k\right)}{\cos\theta \left(k\right)\cos\phi \left(k\right)}$$

It is noted in Equations (30)–(32), $v_{x}\left(k\right)$, $v_{y}\left(k\right)$,$\;v_{z}\left(k\right)$,$\;\omega_{x}\left(k\right)$,$\;\omega_{y}\left(k\right)$ and $\omega_{z}\left(k\right)$ are measured vehicle state values in current time step. ${\hat{x}}_{si}\left(k\right)$ and ${\hat{x}}_{ti}\left(k\right)$ are the estimated suspension spring compression and tyre compression in the current time step. The initial conditions $x_{t0}$ and $x_{s0}$ can be determined by Equations (14) and (26).

The longitudinal and lateral velocity of the unsprung mass in the current time step can be determined based on Equation (24):(37a)$$v_{xui}\left(k\right)=v_{xsi}\left(k\right)-l_{si}\omega_{y}\left(k\right)$$
(37b)$$v_{yui}\left(k\right)=v_{ysi}\left(k\right)+l_{si}\omega_{x}\left(k\right)$$

The velocity on the tyre contact patch in the current time step can be calculated based on Equation (13):(38a)$$v_{xgi}\left(k\right)=\cos\theta \left(k\right)\left(v_{xui}\left(k\right)-\omega_{y}\left(k\right)R_{i}\left(k\right)\right)+\sin\theta \left(k\right)\left({\hat{v}}_{zui}\left(k\right)\cos\phi \left(k\right)+\sin\phi \left(k\right)\left(\omega_{x}\left(k\right)R_{i}\left(k\right)+v_{yui}\left(k\right)\right)\right)$$
(38b)$$v_{ygi}\left(k\right)=\cos\phi \left(k\right)\left(v_{yui}\left(k\right)+\omega_{x}\left(k\right)R_{i}\left(k\right)\right)-{\hat{v}}_{zui}\left(k\right)\sin\phi \left(k\right)$$
where $v_{zui}\left(k\right)$ is hard to measure and can be updated and estimated in every discrete-time iteration with the initial conditions of $v_{zui}\left(0\right)=0$.

The lateral side-slip angle and longitudinal slip ratio of each wheel in discrete time can be determined based on Equations (10)–(12). The lateral tyre force of each wheel $F_{yti}$ can be calculated based on Equation (9). The longitudinal tyre force $F_{xti}$ at the current time step can be directly determined by:(39)$$F_{xti}\left(k\right)=\frac{T_{i}\left(k\right)}{R_{i}\left(k\right)}$$

The tyre forces applied on the wheel $F_{xgi}$ and $F_{ygi}$ can be determined by Equation (23). The tyre force transmitted to vehicle C.G. $F_{xgsi}$ and $F_{ygsi}$ can be determined based on Equation (4). The unsprung mass should be subtracted from forces $F_{xgsi}$ and $F_{ygsi}$:(40a)$$F_{xsi}\left(k\right)=F_{xgsi}\left(k\right)+m_{u}g\sin\theta \left(k\right)$$
(40b)$$F_{ysi}\left(k\right)=F_{ygsi}\left(k\right)-m_{u}g\sin\phi \left(k\right)\cos\theta \left(k\right)$$

Vehicle load transfer of each wheel $F_{dzi}\left(k\right)$ can be obtained from Equation (8). The vertical tyre force $F_{zgi}\left(k\right)$ can be determined based on Equation (14).

The estimated velocity of vehicle suspension in the current time step can be calculated as:(41)$${\hat{\dot{x}}}_{si}\left(k\right)=-v_{zsi}\left(k\right)+{\hat{v}}_{zui}\left(k\right)$$

The estimated velocity of wheel radius change in the current time step can be determined according to Equation (15):(42)$${\hat{\dot{x}}}_{ti}\left(k\right)=v_{xui}\left(k\right)\sin\theta \left(k\right)-\cos\theta \left(k\right)\left({\hat{v}}_{zui}\left(k\right)\cos\phi \left(k\right)+v_{yui}\left(k\right)\sin\phi \left(k\right)\right)$$

The vertical acceleration of unsprung mass in the current time step can be determined according to Equation (22):(43)$$\begin{array}{cc} m_{u}{\hat{\dot{v}}}_{zui}\left(k\right)= & \cos\phi \left(k\right)\left(\cos\theta \left(k\right)\left(F_{zgi}\left(k\right)-m_{u}g\right)+\sin\theta \left(k\right)F_{xgi}\left(k\right)\right)-\sin\phi \left(k\right)F_{ygi}\left(k\right)\\ & -F_{dzi}\left(k\right)-x_{si}\left(k\right)k_{s}-{\dot{x}}_{si}\left(k\right)b_{s}-m_{u}\left({\hat{v}}_{zui}\left(k\right)\omega_{x}\left(k\right)-v_{xui}\left(k\right)\omega_{y}\left(k\right)\right) \end{array}$$

Finally, the estimated values of ${\hat{x}}_{si}$, ${\hat{x}}_{ti}$ and ${\hat{v}}_{zui}$ in the next time step can be estimated by:(44a)$${\hat{x}}_{si}\left(k+1\right)={\hat{x}}_{si}\left(k\right)+{\dot{x}}_{si}\left(k\right)\left(t\left(k+1\right)-t\left(k\right)\right)$$
(44b)$${\hat{x}}_{ti}\left(k+1\right)={\hat{x}}_{ti}\left(k\right)+{\dot{x}}_{ti}\left(k\right)\left(t\left(k+1\right)-t\left(k\right)\right)$$
(44c)$${\hat{v}}_{zui}\left(k+1\right)={\hat{v}}_{zui}\left(k\right)+{\dot{v}}_{zui}\left(k\right)\left(t\left(k+1\right)-t\left(k\right)\right)$$

The predicted vehicle state values of $v_{x},v_{y},\omega_{z},\phi$ in the next n time steps can be determined by following equations:(45a)$${\hat{v}}_{x}\left(k+n\right)=v_{x}\left(k+n-1\right)+{\dot{v}}_{x}\left(k\right)\left(t\left(k+n\right)-t\left(k+n-1\right)\right)$$
(45b)$${\hat{v}}_{y}\left(k+n\right)=v_{y}\left(k+n-1\right)+{\dot{v}}_{y}\left(k\right)\left(t\left(k+n\right)-t\left(k+n-1\right)\right)$$
(45c)$${\hat{\omega }}_{z}\left(k+n\right)=\omega_{z}\left(k+n-1\right)+{\dot{\omega }}_{z}\left(k\right)\left(t\left(k+n\right)-t\left(k+n-1\right)\right)$$
(45d)$$\begin{array}{cc} \hat{\phi }\left(k+n\right)= & \phi \left(k+n-1\right)\\ & \;+(\omega_{x}\left(k\right)+\omega_{y}\left(k\right)\sin\left(\phi \left(k+n-1\right)\right)\tan\left(\theta \left(k\right)\right)\\ & \;+\omega_{z}\left(k\right)\cos\left(\phi \left(k+n-1\right)\right)\tan\left(\theta \left(k\right)\right)) \end{array}$$
where $n=1,2,\ldots ,n_{p}$, $n_{p}$ presents the predicted horizontal of the state predictor. It is noted that in a relatively short prediction time, the acceleration values ${\dot{v}}_{x}$, ${\dot{v}}_{y}$, ${\dot{\omega }}_{z}$ can be assumed as constant values and the vehicle states estimated by Equation (45) within a small, predicted horizontal can have acceptable prediction performance.

### 4.2. The Upper-Level Control Mode Supervisor

Based on the predicted vehicle states from the vehicle’s future state predictor and the diagram of the vehicle stability region determined in Figure 5 and Figure 6, the upper-level control supervisor can determine the best suitable control mode from cruise control mode, yaw-plane stability control mode and roll-stability control mode.

The cruise control mode only aims to maintain the desired longitudinal velocity. In the yaw-plane stability control mode, the desired yaw rate and body side-slip angle can be achieved. In the roll-stability control mode, the vehicle roll stability can be improved and rollover can be prevented.

The control mode selection rules can be presented as follows, and are also illustrated in Figure 8:

(1)Determine the predicted state values of lateral velocity ${\hat{v}}_{y}\left(k+n\right)$, yaw rate $\hat{r}\left(k+n\right)$ and the predicted load transfer ratio $\hat{R}\left(k+n\right)$. The load transfer ratio can be presented by the following equation, according to [25]:(46)$$\hat{R}\left(k+n\right)=\frac{2K_{\phi }\phi \left(k+n\right)+2C_{\phi }\dot{\phi }\left(k+n\right)}{cmg}$$(2)According to the predicted vehicle lateral velocity ${\hat{v}}_{y}\left(k+n\right)$, yaw rate $\hat{r}\left(k+n\right)$ and diagram of yaw stability region (as in Figure 6 and Figure 7), if the vehicle is moving outside the yaw stability region, the vehicle’s yaw-plane stability control mode is selected.(3)According to the predicted value of load transfer ratio (LTR) $\hat{R}\left(k+n\right)$, if $\hat{R}\left(k+n\right)<0.2$, the rollover is unlikely to happen and the roll-stability control mode is disabled; if $0.2\leq \hat{R}\left(k+n\right)\leq 0.6$, the vehicle is likely to rollover and the roll-stability control mode is selected. These threshold values are determined according to [28].(4)If the driver wants to maintain the desired longitudinal velocity, the cruise control mode is selected with full longitudinal velocity control. It is noted the yaw-plane stability control mode, the roll-stability control mode and the cruise control mode could be activated at the same time when their active threshold conditions are satisfied.(5)When $\hat{R}\left(k+n\right)>0.6$ and the vehicle is in the critical roll-stability mode, the cruise control and yaw-plane stability control is disabled and the vehicle is in a full brake. According to [28], during the critical roll-stability mode, the inside wheels of the vehicle may have already lifted off and the vehicle may roll over immediately. Rollover prevention is far more important than yaw stability. Therefore, the full brake strategy is selected for critical roll-stability mode by neglecting other control targets.

### 4.3. The Lower-Level 4 DOF MPC Algorithm

**Assumption** **2.**
*It is assumed that the vehicle states*

$x_{si}\left(k\right)$

*,*

$x_{ti}\left(k\right)$

*and*

$v_{zui}\left(k\right)$

*are all assumed to be successfully estimated by the proposed state estimator. The vehicle longitudinal velocity, lateral velocity, vertical velocity, yaw angle, roll angle, pitch angle, yaw rate, roll rate and pitch rate are assumed to be easily measured or estimated. In addition, the sideslip angle and slip ratio of the individual wheels are assumed to be known.*


The cost function of the proposed 4 DOF MPC can be presented as the following equation:(47)$$\begin{array}{cc} \underset{T_{j}\left(\text{healthy wheels}\right)}{\min}J & \\ & =\sum_{i=1}^{N}[a_{1}{\left({\hat{v}}_{x}\left(k+i\right)-v_{xd}\left(k+i\right)\right)}^{2}+a_{2}{\left(\hat{\beta }\left(k+i\right)-\beta_{d}\left(k+i\right)\right)}^{2}\\ & +a_{3}{\left({\hat{\omega }}_{z}\left(k+i\right)-\omega_{zd}\left(k+i\right)\right)}^{2}+a_{4}\hat{R}{\left(k+i\right)}^{2}]\\ & +[a_{1}{\left({\hat{v}}_{x}\left(k+N+1\right)-v_{xd}\left(k+N+1\right)\right)}^{2}+a_{2}{\left(\hat{\beta }\left(k+N+1\right)-\beta_{d}\left(k+N+1\right)\right)}^{2}\\ & +a_{3}{\left({\hat{\omega }}_{z}\left(k+N+1\right)-\omega_{zd}\left(k+N+1\right)\right)}^{2}+a_{4}\hat{R}{\left(k+N+1\right)}^{2}] \end{array}$$
where $v_{xd}$ is the desired longitudinal velocity. It is noted that after the tyre deflation, the allocated braking or traction torque on the deflated tyre will further deteriorate the vehicle stability. Therefore, the optimization algorithm (47) only allocates the individual wheel torque $T_{i}$ to healthy wheels.

$\beta_{d}$ and $\omega_{zd}$ are desired side-slip angle of C.G. and desired yaw rate, which are determined by a 2 DOF desired vehicle model:(48a)$${\dot{\beta }}_{d}=\left\{\left(C_{\alpha }+C_{\alpha }\right)\delta -2\beta_{d}\left(C_{\alpha }+C_{\alpha }\right)-\left[mv_{d}\omega_{zd}+\frac{2\left(l_{f}C_{\alpha }-l_{r}C_{\alpha }\right)\omega_{zd}}{v_{d}}\right]\right\}\frac{1}{mv_{d}}$$
(48b)$${\dot{\omega }}_{zd}=\left\{2l_{f}C_{\alpha }\delta -2\left(l_{f}C_{\alpha }-l_{r}C_{\alpha }\right)\beta_{d}-\frac{2\omega_{zd}\left(l_{f}^{2}C_{\alpha }+l_{r}^{2}C_{\alpha }\right)}{v_{d}}\right\}\frac{1}{J_{z}}$$

It is noted that the desired yaw rate cannot exceed the maximum yaw rate:(49)$$\omega_{zd}=\underset{}{\min}\left(\omega_{zd},\frac{\mu g}{v_{xd}}\right)$$

The scaling factors $a_{1},a_{2},a_{3},a_{4}$ can be adjusted to reflect different control modes: (1) when it is required to disable the cruise control mode, $a_{1}=0$; (2) when it is required to disable the yaw stability control mode, $a_{2}=a_{3}=0$; (3) when it is required to disable the roll-stability control, $a_{4}=0$. It is noted that in order to progressively disable different modes, the function tanhx is applied.
(50a)$$a_{1}=\tanh x_{1}b_{1}$$
(50b)$$a_{2}=\tanh x_{2}b_{2}$$
(50c)$$a_{3}=\tanh x_{3}b_{3}$$
(50d)$$a_{4}=\tanh x_{4}b_{4}$$

When x→+∞, tanhx→1; When x→0, tanhx→0. $x_{1-4}$ is related to the evaluation criteria of the different mode selections. $b_{1-4}$ is the weighting factors of each individual term.

In optimization cost function (47), the longitudinal velocity $v_{x}$, side-slip angle in C.G., yaw rate $\omega_{z}$ and roll angle ϕ can be predicted by the following equations:(51a)$${\hat{v}}_{x}\left(k+1\right)=v_{x}\left(k\right)+{\dot{v}}_{x}\left(k\right)\Delta t$$
(51b)$${\hat{v}}_{y}\left(k+1\right)=v_{y}\left(k\right)+{\dot{v}}_{y}\left(k\right)\Delta t$$
(51c)$${\hat{\omega }}_{z}\left(k+1\right)=\omega_{z}\left(k\right)+{\dot{\omega }}_{z}\left(k\right)\Delta t$$
(51d)$$\hat{\phi }\left(k+1\right)=\phi \left(k\right)+\left(\omega_{x}+\omega_{z}\left(k\right)\theta \right)\Delta t$$
(51e)$$\hat{\beta }\left(k+1\right)=\tan^{-1}\frac{{\hat{v}}_{y}\left(k+1\right)}{v_{x}\left(k+1\right)}$$
where ${\dot{v}}_{x}\left(k\right)$, ${\dot{v}}_{y}\left(k\right)$, ${\dot{\omega }}_{z}\left(k\right)$ can be determined based on Equation (1):(52a)$${\dot{v}}_{x}\left(k\right)=\omega_{y}v_{z}+\omega_{z}v_{y}\left(k\right)+\frac{\sum_{i=1,2,3,4}F_{xsi}}{m}+g\sin\theta$$
(52b)$${\dot{v}}_{y}\left(k\right)=\omega_{z}\left(k\right)v_{z}-\omega_{z}\left(k\right)v_{x}\left(k\right)+\frac{\sum_{i=1,2,3,4}F_{ysi}}{m}-g\sin\phi \cos\theta$$
(52c)$${\dot{\omega }}_{z}\left(k\right)=\frac{l_{f}\left(F_{ys1}+F_{ys2}\right)-l_{r}\left(F_{ys3}+F_{ys4}\right)}{J_{z}}+\frac{c\left(-F_{xs1}+F_{xs2}-F_{xs3}+F_{xs4}\right)}{2J_{z}}$$
where $F_{xsi}$ and $F_{ysi}$ can be determined by Equations (32)–(40) and estimated vehicle states ${\hat{x}}_{si}\left(k\right)$, ${\hat{x}}_{ti}\left(k\right)$ and ${\hat{v}}_{zui}\left(k\right)$.

Therefore, according to Equations (51) and (52), the cost function (47) of MPC can be clearly rewritten as the equation below:(53)$$\begin{array}{cc} \underset{T_{j}\left(\text{healthy wheels}\right)}{\min}J & =\sum_{i=1}^{N}\{a_{1}{\left[v_{x}\left(k+i\right)+\left(\left(\underset{i=1,2,3,4}{\sum }\cos\theta \frac{T_{i}\left(k+i-1\right)}{R_{i}m}\right)+A_{1}\right)\Delta t-v_{xd}\left(k+i\right)\right]}^{2}\\ & +a_{2}{\left[\tan^{-1}\frac{v_{y}\left(k+i\right)+\left(\left(\sum_{i=1,2,3,4}\sin\theta \sin\phi \frac{T_{i}\left(k+i-1\right)}{R_{i}m}\right)+A_{2}\right)\Delta t}{v_{x}\left(k+i\right)+\left(\left(\sum_{i=1,2,3,4}\cos\theta \frac{T_{i}\left(k+i-1\right)}{R_{i}m}\right)+A_{1}\right)\Delta t}-\beta_{d}\left(k+i\right)\right]}^{2}\\ & +a_{3}{\left[\omega_{z}\left(k+i\right)+\left(\left(\frac{c\cos\theta }{2J_{z}}\left(T_{2}-T_{1}+T_{4}-T_{3}\right)\right)+A_{3}\right)\Delta t-\omega_{zd}\left(k+i\right)\right]}^{2}\\ & +a_{4}{\left[\frac{2k_{\phi }\left(\phi \left(k+i\right)+\left(\omega_{x}+\omega_{z}\left(k+i\right)\right)\Delta t\right)+c_{\phi }\left(\omega_{x}+\omega_{z}\left(k+i\right)\right)}{cmg}\right]}^{2}\}\\ & +[a_{1}{\left[v_{x}\left(k+N+1\right)+\left(\left(\underset{i=1,2,3,4}{\sum }\cos\theta \frac{T_{i}\left(k+N\right)}{R_{i}m}\right)+A_{1}\right)\Delta t-v_{xd}\left(k+N+1\right)\right]}^{2}\\ & +a_{2}{\left[\tan^{-1}\frac{v_{y}\left(k+N+1\right)+\left(\left(\sum_{i=1,2,3,4}\sin\theta \sin\phi \frac{T_{i}\left(k+N\right)}{R_{i}m}\right)+A_{2}\right)\Delta t}{v_{x}\left(k+N+1\right)+\left(\left(\sum_{i=1,2,3,4}\cos\theta \frac{T_{i}\left(k+N\right)}{R_{i}m}\right)+A_{1}\right)\Delta t}-\beta_{d}\left(k+N+1\right)\right]}^{2}\\ & +a_{3}{\left[\omega_{z}\left(k+N+1\right)+\left(\left(\frac{c\cos\theta }{2J_{z}}\left(T_{2}-T_{1}+T_{4}-T_{3}\right)\right)+A_{3}\right)\Delta t-\omega_{zd}\left(k+N+1\right)\right]}^{2}\\ & +a_{4}{\left[\frac{2k_{\phi }\left(\phi \left(k+N+1\right)+\left(\omega_{x}+\omega_{z}\left(k+N+1\right)\right)\Delta t\right)+c_{\phi }\left(\omega_{x}+\omega_{z}\left(k+N+1\right)\right)}{cmg}\right]}^{2}] \end{array}$$
where $A_{1}=\omega_{y}v_{z}+\omega_{z}\left(k\right)v_{y}\left(k\right)+g\sin\theta +\underset{i=1,2,3,4}{\sum }\frac{-\sin\theta F_{zgi}+m_{u}g\sin\theta }{m}$, $A_{2}=\omega_{z}\left(k\right)v_{z}-\omega_{z}\left(k\right)v_{x}-g\sin\phi \cos\theta +\underset{i=1,2,3,4}{\sum }\frac{\cos\phi F_{ygi}+\sin\phi \cos\theta F_{zgi}-m_{u}g\sin\phi \cos\theta }{m}$, $A_{3}=\frac{l_{f}\left(F_{ys1}+F_{ys2}\right)-l_{r}\left(F_{ys3}+F_{ys4}\right)}{J_{z}}+\frac{c\sin\theta \left(F_{zg1}-F_{zg2}+F_{zg3}-F_{zg4}\right)}{2J_{z}}$.

The stability proof of the proposed MPC controller is presented in the Appendix A.

## 5. Simulation Results

In this section, the proposed 14 DOF MPC is implemented on the simulation platform of Matlab Simulink to present the combined yaw-plane stability and roll-stability control performance. Furthermore, in order to do the comparative study and show the advantages of the proposed 14 DOF MPC, the control performance of the traditional 8 DOF MPC is also presented. This 8 DOF model considers the longitudinal motion, lateral motion, yaw motion, roll motion and rotational motion of four wheels and includes the yaw-plane stability control mode and roll-stability control mode. If R<0.6, the combined yaw-plane stability control mode and roll-stability control mode is enabled; if R≥0.6, only the roll-stability control mode is enabled, and the vehicle has the full brake.

Three sets of simulation results are presented in the following paragraphs: in the first set of simulations, the proposed MPC is working under the normal driving mode and the yaw-plane stability is the focus; in the second and third sets of simulations, the proposed MPC is under the yaw and roll-stability control mode and the roll stability is the focus. The sampling time of the proposed 14 DOF MPC and 8 DOF MPC was 0.005 s and the prediction horizon was five steps. Due to the large computational effort, the MPC sampling time was chosen as 0.005 s, which is the same as the sampling time constant of vehicle plant dynamics model. The control horizon is also five steps.

In the first set of simulations, the vehicle is assumed to move along the straight line with an initial longitudinal velocity of 40 m/s. The tyre–road friction coefficient is assumed as 0.9. Tyre blow-out happens at the front left wheel after 2 s and the changing of the front-left tyre parameter after tyre blow-out is shown in Figure 9. Figure 10 compares the vehicle dynamics responses when the proposed 14 DOF MPC and traditional 8 DOF MPC are applied. Figure 10f presents the changed real-time scaling factors of the optimization cost function of 14 DOF MPC determined by the upper-level control mode supervisor, which shows that 14 DOF MPC only chooses the cruise control mode 2 s before, when all the tyres are in a healthy condition. After 2 s, the 14 DOF MPC switches into combined cruise control mode and yaw stability control mode. Figure 10a shows the similar control performances of longitudinal velocity for both of the two methods. Figure 10b,c prove that the proposed 14 DOF MPC has a better yaw rate and body side-slip angle control performance than 8 DOF MPC. The 8 DOF MPC has a larger over-shoot of yaw rate and body side-slip angle response after tyre blow-out at 2 s than the no controller applied condition. According to Figure 10d,e, since the LTR is less than the roll-stability control threshold value of 0.2, the roll-stability control is disabled and the proposed 14 DOF MPC cannot control the roll angle and the value of LTR. The motor control torques of the different controllers are shown in Figure 11.

Figure 12 shows the sensitivity analysis of the proposed 14 DOF MPC under different tyre–road friction coefficient conditions and different rolling resistance coefficients after a tyre blow-out. The proposed 14 DOF MPC controller shows a quite robust control performance.

In the second set of simulations, the vehicle started to have the J-turn motion after 2 s (the input steering angle is shown in Figure 13). The initial longitudinal velocity was 40 m/s and tyre–road friction coefficient was 0.9. After 5 s, the front left tyre blows out and the changed tyre parameters are shown in Figure 14. Figure 15 presents and compares the vehicle dynamics responses of the proposed 14 DOF MPC and traditional 8 DOF MPC. Figure 15f shows that 14 DOF MPC switches from the pure cruise control mode into the combined cruise control and yaw stability control mode after the beginning of the J-turn in 2 s, then switches into combined cruise control mode, roll-stability and yaw-stability control mode after 7 s. Figure 15a–c all prove that the proposed 14 DOF MPC can significantly improve the longitudinal velocity, yaw rate and body side-slip angle response after a tyre blow-out compared with 8 DOF MPC. According to Figure 15d,e, after 5 s, the dynamics responses of LTR and roll angle of 14 DOF MPC are improved because the roll-stability control mode is enabled according to Figure 15f. The motor control torques of the different controllers are shown in Figure 16. Figure 17 suggests the control performance of 14 DOF MPC when considering the measurement noise (measured yaw rate with white noise variance of 0.02 rad/s) and shows good robustness on measurement noise.

In the third set of simulations, the fishhook steering manoeuvre (shown in Figure 18) was applied to test the control performance of the proposed method. The initial longitudinal velocity and tyre–road friction coefficient were the same as the second set of simulations. After 5 s, the front left tyre blows out and the changed tyre parameters are the same as in Figure 14. Figure 19 presents and compares the dynamics performance of the proposed 14 DOF MPC and traditional 8 DOF MPC. Figure 19f shows that after 3 s, the control mode of 14 DOF MPC switches from pure cruise control mode into the combined cruise control, yaw stability and roll-stability control mode. Figure 19b,c shows that the proposed 14 DOF MPC and 8 DOF MPC cannot achieve the desired yaw rate and side-slip angle. Figure 19d,e prove that the proposed 14 DOF MPC has much better roll-stability control performance than 8 DOF MPC. In all three sets of simulations, the yaw-stability and roll-stability dynamics control performance of 8 DOF MPC was significantly compromised. This is mainly because the 8 DOF MPC is based on the 8 DOF vehicle dynamics model which cannot accurately present the vehicle dynamics performance during tyre blow-out. The motor control torques of different controllers are shown in Figure 20.

## 6. Conclusions

This study first proposes a comprehensive 14 DOF vehicle dynamics model to describe the vehicle dynamics performance after a tyre blow-out. Then, based on the proposed 14 DOF model, a non-linear coordinate control framework based on MPC is proposed. The simulation results can be summarised as follows:(1)The proposed 14 DOF vehicle dynamics model can successfully describe the effect of the changed tyre vertical stiffness, cornering stiffness and rolling resistance after a tyre blow-out on the vehicle dynamics performance.(2)The proposed vehicle state predictor can successfully predict the vehicle’s future states and the proposed upper-level control mode supervisor can use the predicted vehicle states to select the suitable control mode.(3)The proposed lower-level MPC based on the 14 DOF model can successfully improve the vehicle yaw-dynamics performance including the yaw rate and side-slip angle in the scenarios of tyre blow-out during straight line moving and J-turn manoeuvre.(4)The proposed lower-level MPC based on the 14 DOF model can successfully improve the roll stability in the challenging scenario of a tyre blow-out during a fishhook manoeuvre when the vehicle has a big load transfer.(5)The traditional MPC based on the 8 DOF model cannot successfully improve the vehicle yaw stability and roll stability of the vehicle after tyre blow-out.

In the future, the effect of tyre blow-out on the autonomous steering system of autonomous vehicles will be investigated and the design of a fault-tolerant steering control strategy to overcome the issue of tyre blow-out will be focused on.

## Figures and Tables

**Figure 1 sensors-21-08328-f001:**
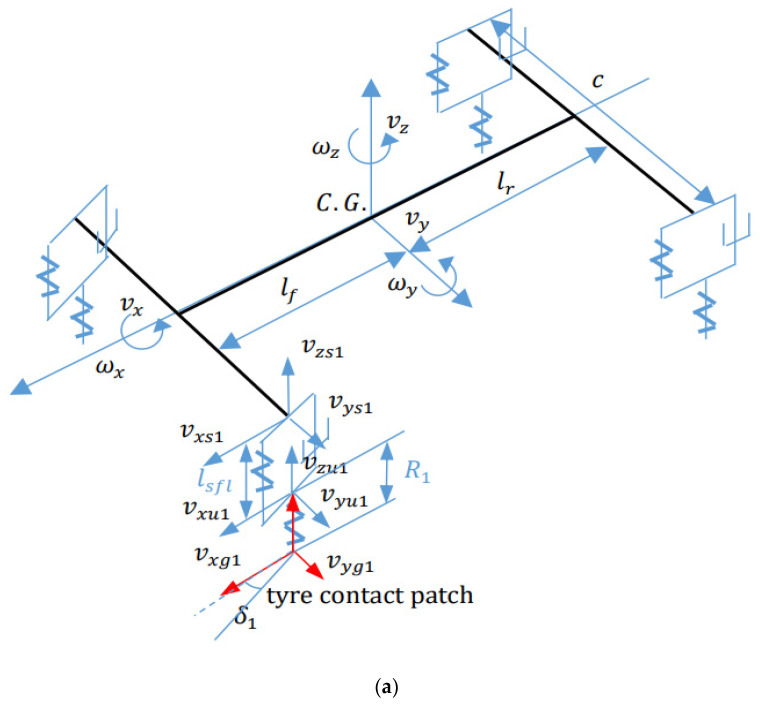
Schematic of 14 DOF vehicle dynamics model (**a**) yaw plane (**b**) roll plane.

**Figure 2 sensors-21-08328-f002:**
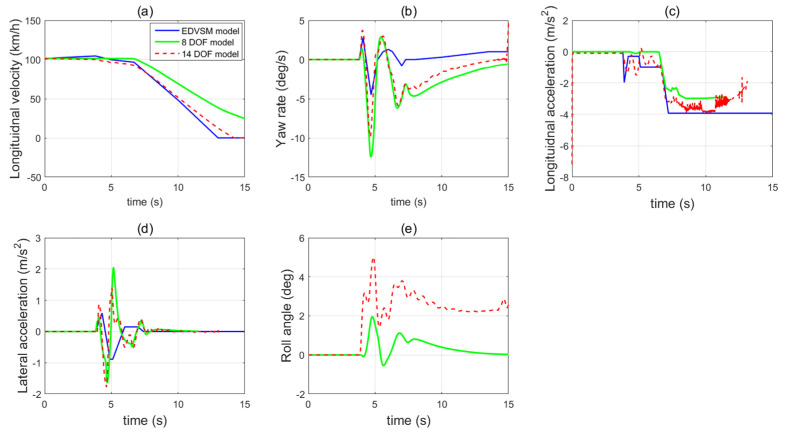
The vehicle state dynamics responses of 8 DOF model and 14 DOF model after tyre deflation (**a**) longitudinal velocity (**b**) yaw rate (**c**) longitudinal acceleration (**d**) lateral acceleration (**e**) roll angle.

**Figure 3 sensors-21-08328-f003:**
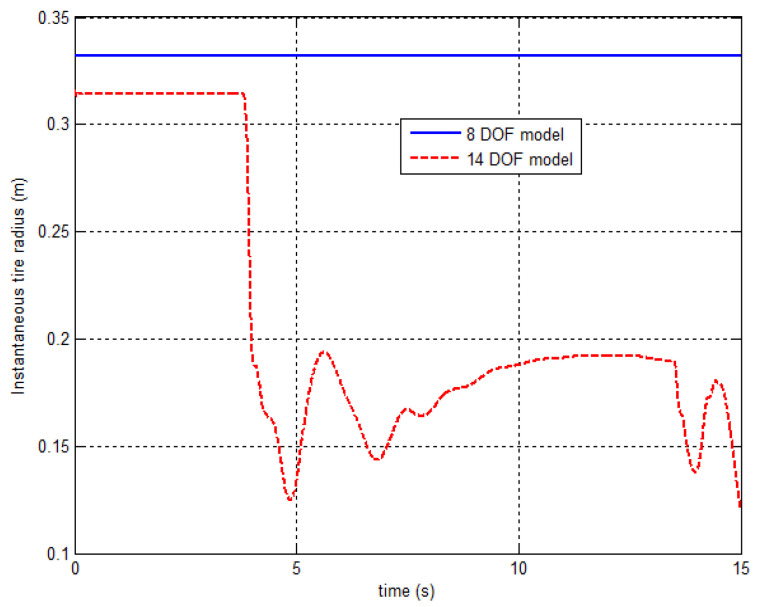
The tyre’s instantaneous radius responses of 8 DOF model and 14 DOF model.

**Figure 4 sensors-21-08328-f004:**
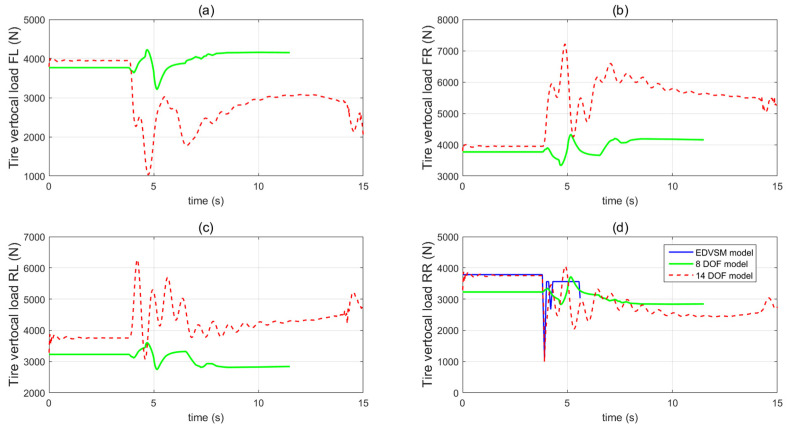
The vertical load responses of 8 DOF model and 14 DOF model (**a**) front left wheel (**b**) front right wheel (**c**) rear left wheel (**d**) rear right wheel.

**Figure 5 sensors-21-08328-f005:**
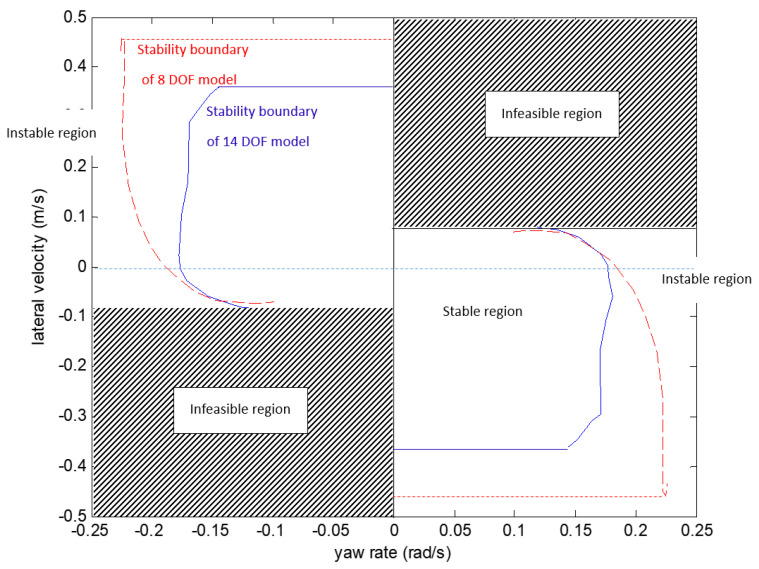
Compares the yaw-plane stability region of 14 DOF model and 8 DOF model ($v_{x}=20,\;\mu =1$).

**Figure 6 sensors-21-08328-f006:**
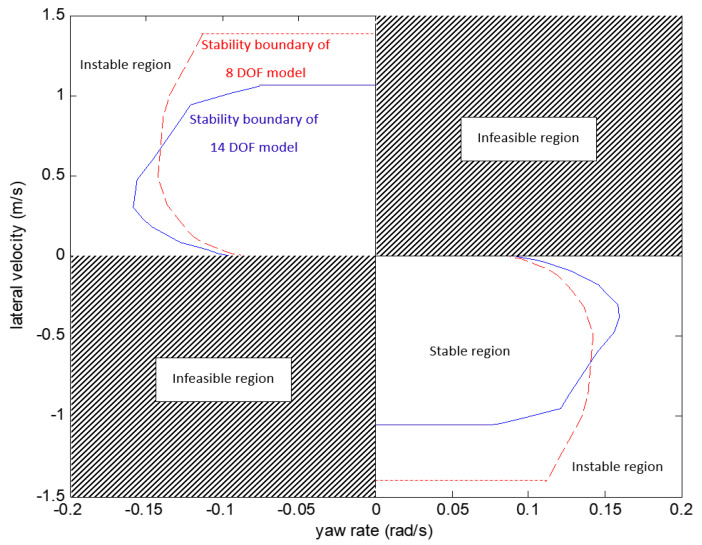
Compares the yaw-plane stability region of 14 DOF model and 8 DOF model ($v_{x}=40,\;\mu =1$).

**Figure 7 sensors-21-08328-f007:**
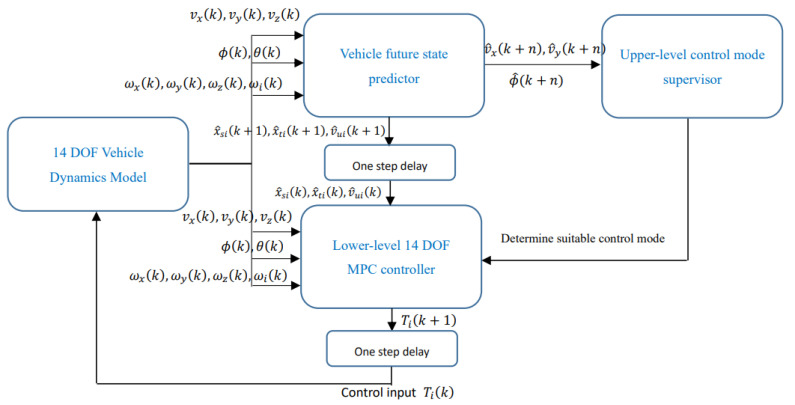
The structure diagram of proposed three-dimensional integrated stability control framework.

**Figure 8 sensors-21-08328-f008:**
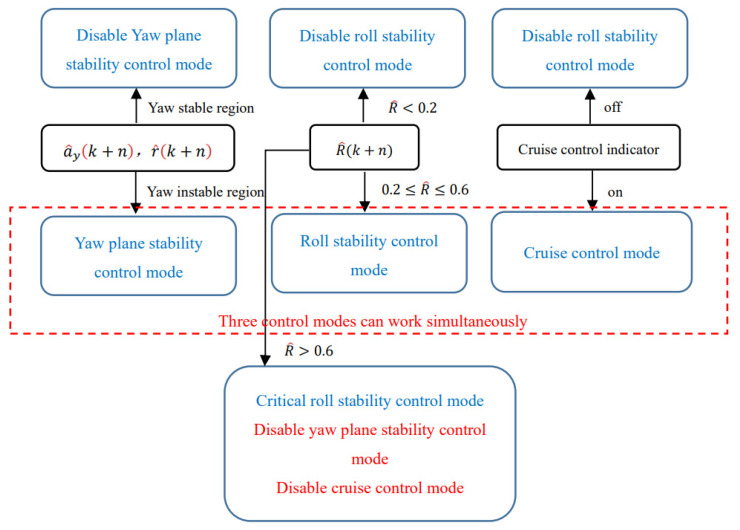
The diagram of the proposed control mode selection rules.

**Figure 9 sensors-21-08328-f009:**
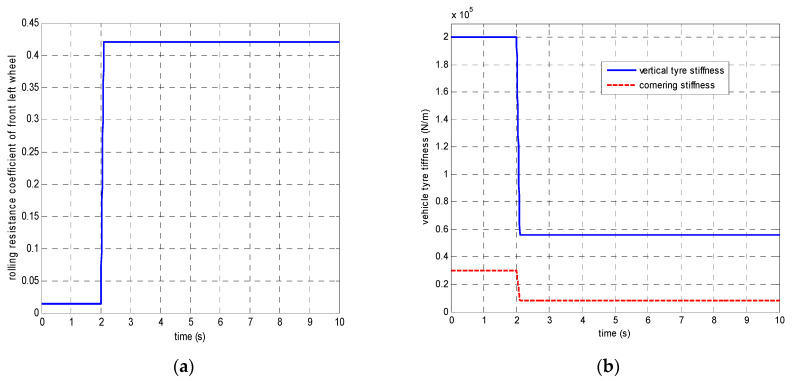
The changing of front-left tyre parameter after tyre blow-out (**a**) rolling resistance coefficient (**b**) tyre stiffness.

**Figure 10 sensors-21-08328-f010:**
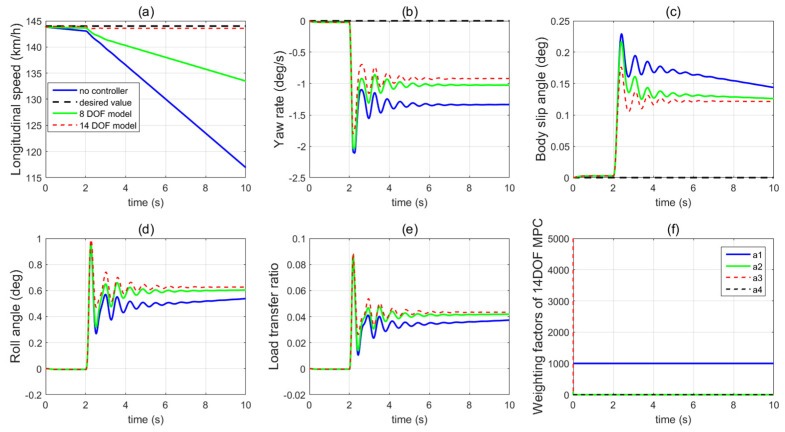
Vehicle dynamics performance when proposed controller applied in the first set of simulations (**a**) longitudinal velocity, (**b**) yaw rate, (**c**) body side-slip angle, (**d**) roll angle, (**e**) LTR, (**f**) scaling factors of 14 DOF MPC.

**Figure 11 sensors-21-08328-f011:**
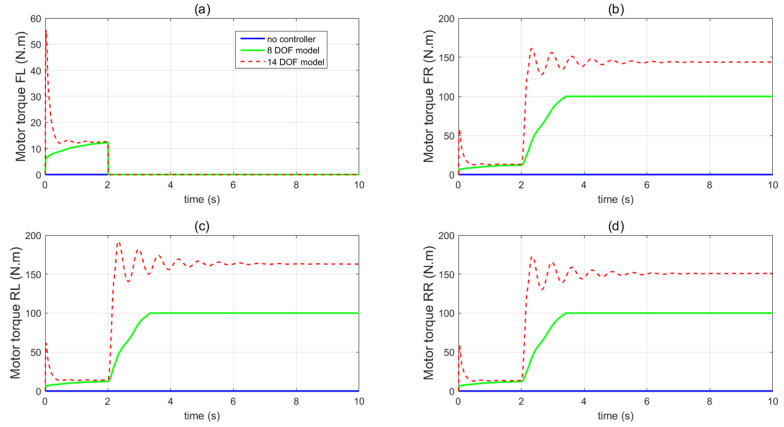
Vehicle motor control inputs in the first set of simulations (**a**) front left wheel, (**b**) front right wheel, (**c**) rear left wheel, (**d**) rear right wheel.

**Figure 12 sensors-21-08328-f012:**
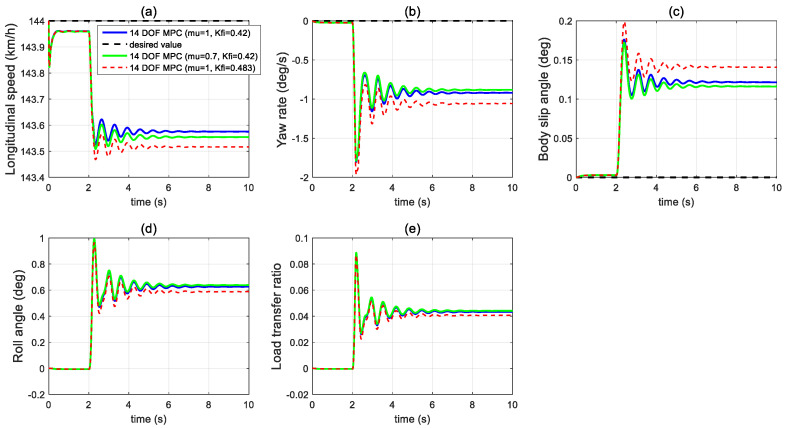
Sensitivity analysis of the dynamics performance of 14 DOF MPC in the first set of simulations (**a**) longitudinal velocity, (**b**) yaw rate, (**c**) body side-slip angle, (**d**) roll angle, (**e**) LTR.

**Figure 13 sensors-21-08328-f013:**
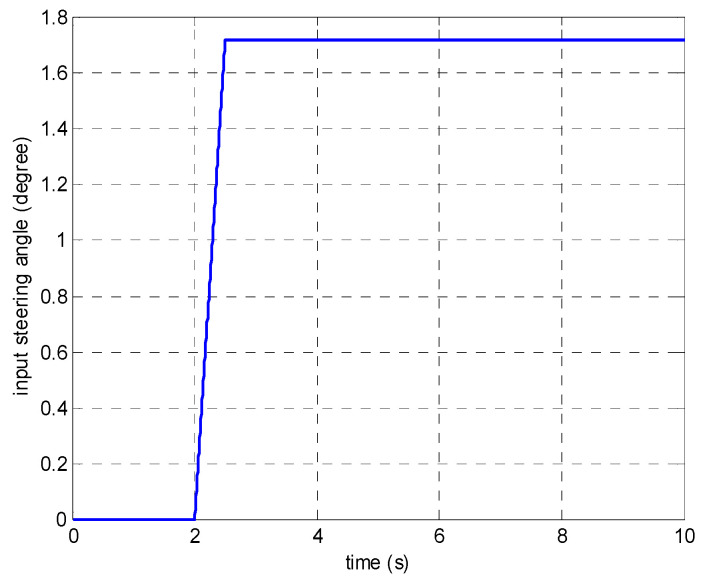
Driver’s input steering angle in the second set of simulations.

**Figure 14 sensors-21-08328-f014:**
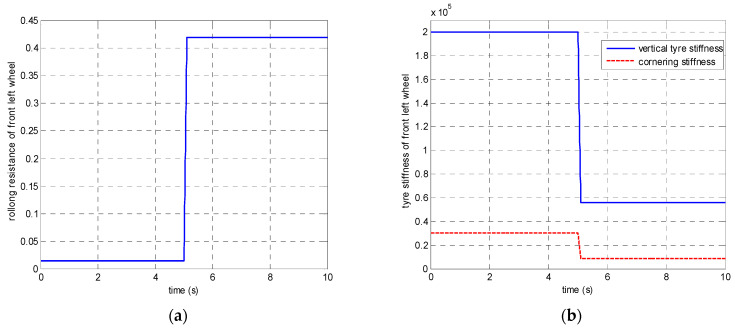
The changing of front-left tyre parameters after tyre blow-out in the second set of simulations (**a**) rolling resistance, (**b**) tyre vertical and cornering stiffness.

**Figure 15 sensors-21-08328-f015:**
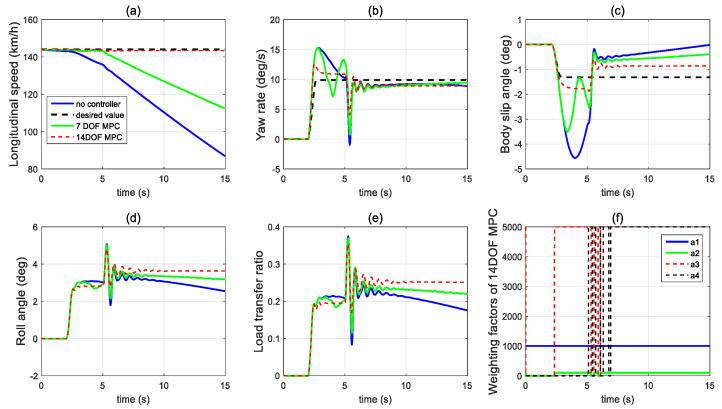
Vehicle dynamics performance when proposed controller applied in the second set of simulations (**a**) longitudinal velocity, (**b**) yaw rate, (**c**) body side-slip angle, (**d**) LTR, (**e**) roll angle, (**f**) scaling factors of 14 DOF MPC.

**Figure 16 sensors-21-08328-f016:**
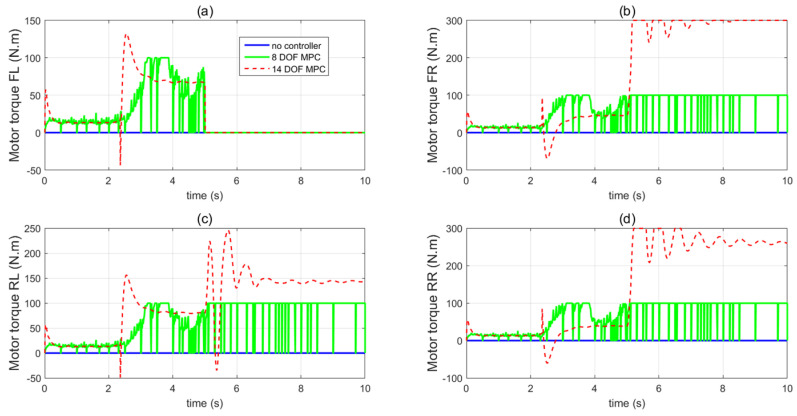
Vehicle motor control inputs in the second set of simulations (**a**) front left wheel, (**b**) front right wheel, (**c**) rear left wheel, (**d**) rear right wheel.

**Figure 17 sensors-21-08328-f017:**
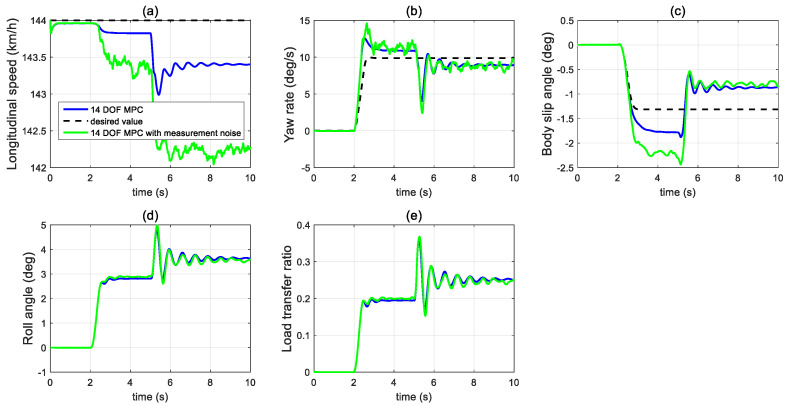
Vehicle dynamics performance when considering the measurement noise in the second set of simulations (**a**) longitudinal velocity, (**b**) yaw rate, (**c**) body side-slip angle, (**d**) LTR, (**e**) roll angle.

**Figure 18 sensors-21-08328-f018:**
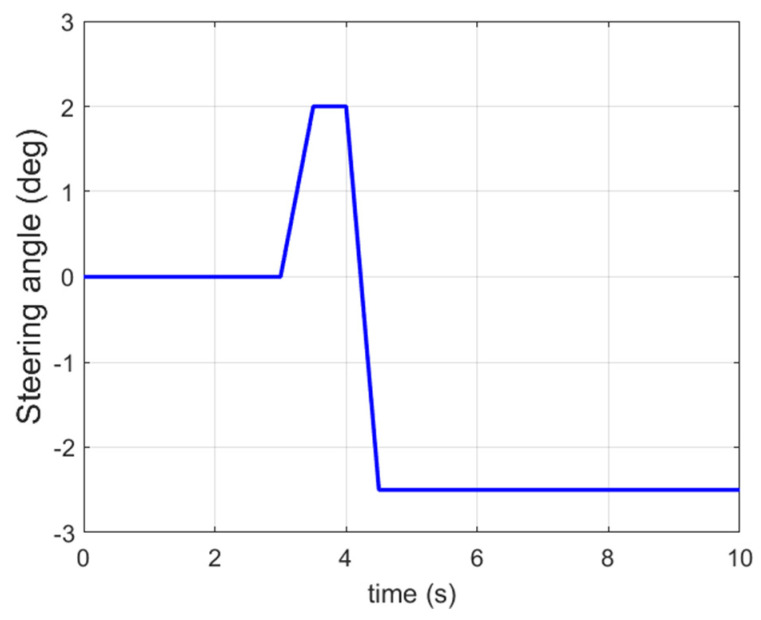
Input steering angle in the third set of simulations.

**Figure 19 sensors-21-08328-f019:**
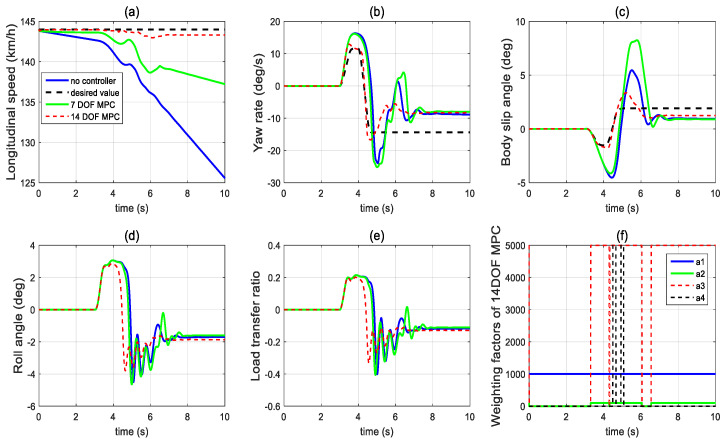
Vehicle dynamics performance when proposed controller applied in the third set of simulations (**a**) longitudinal velocity, (**b**) yaw rate, (**c**) body side-slip angle, (**d**) roll angle, (**e**) LTR, (**f**) scaling factors of 14 DOF MPC.

**Figure 20 sensors-21-08328-f020:**
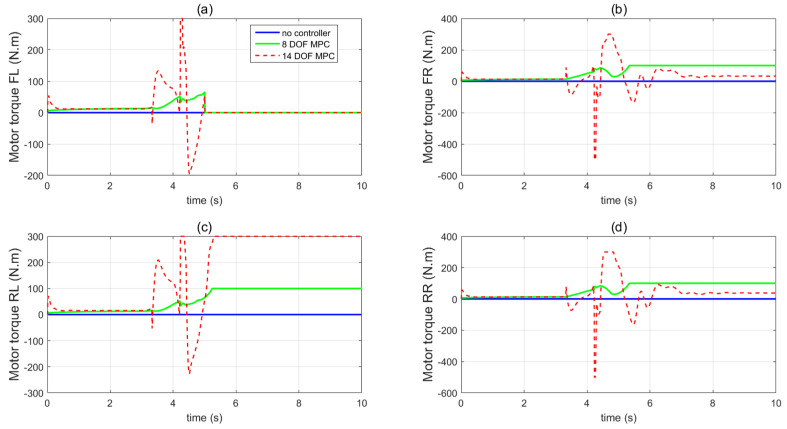
Vehicle motor control inputs in the third set of simulations (**a**) front left wheel, (**b**) front right wheel, (**c**) rear left wheel, (**d**) rear right wheel.

**Table 1 sensors-21-08328-t001:** Vehicle parameters of 8 DOF model and 14 DOF model (same as [7]).

Vehicle Mass	m	1440 kg
Distance between front axle and C.G.	$l_{f}$	1.016 m
Distance between rear axle and C.G.	$l_{r}$	1.524 m
Track width	b	1.5 m
Pitch moment of inertia	$J_{x}$	900 kg.m^2^
Roll moment of inertia	$J_{y}$	900 kg.m^2^
Yaw moment of inertia	$J_{z}$	2000 kg.m^2^
Height of C.G.	h	0.75 m
Front suspension stiffness	$k_{sf}$	35,000 N/m
Rear suspension stiffness	$k_{sr}$	35,000 N/m
Front suspension damping ratio	$b_{sf}$	2500 N.s/m
Rear suspension damping ratio	$b_{sr}$	2500 N.s/m
Vertical front tyre stiffness	$k_{tf}$	200,000 N/m
Vertical rear tyre stiffness	$k_{tr}$	200,000 N/m
Tyre cornering stiffness	$C_{\alpha }$	30,000 N/m

## Data Availability

Not applicable.

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
