# Peer review of "A Three-Dimensional Integrated Non-Linear Coordinate Control Framework for Combined Yaw- and Roll-Stability Control during Tyre Blow-Out"

_sensors, 2021, doi:10.3390/s21248328_

Round 1

Reviewer 1 Report

The presented topic can be considered quite interesting and useful to study the impact of the abrupt change in rolling resistance, cornering stiffness and vertical stiffness after tyre blowout on vehicle dynamic performance.

The paper is well organised and the arguments are convincingly presented, although more detail could be provided regarding the sensitivity on the results of the rolling resistance assumptions.

The authors state that "the typical rolling resistance stiffness for the light vehicle is around 0.012 and 0.015   and it is argued in [7] that this value increases thirty times  after the tyre blowout". Therefore, in the paper, the rolling resistance coefficient after tyre burst is assumed to be 0.42.

The resistance value proposed in [7] is an average value obtained from validated models of different vehicles. It could be interesting to evaluate the sensitivity of this value on the simulation results using different values (e.g. + or - 15% of 0.42).

Reviewer 2 Report

This paper proposes a three-dimensional integrated nonlinear coordinate control strategy for combined yaw and roll stability control during tire blow-out. The comments are listed:

1) Some latest researches are required to add in distributed drive electric vehicles. It is recommend but not confined: “10.1109/TTE.2021.3085849”.

2) To highlight, the contributions should be listed in the form of points in the introduction, rather than combined with work labors.

3) What’s the meaning of “c” in eq. 28? It is recommended to add a list of abbreviation at the beginning of the paper.

4) In assumption 1, the references should be added.

5) The symbol of carriage return will influence the beauty. It should not show in fig. 8.

6) As the accuracy of model has been verified, why not apply the model in EDVSM to validate your strategy? It is more general to adopt a high-fidelity model as validation model to test, for better persuasion.

7) In validations, other methods, cycles, tire-road friction factors, should be added for comprehension.

8) How to consider the buffeting when modes change in your strategy?

Reviewer 3 Report

This manuscript has proposed an interesting topic about the integrated coordinate controller design for the vehicle roll and yaw motion during tyre blow-out scenario. The complex 14 DOF vehicle dynamics model is applied to simulate the vehicle performance during tyre blow-out and is also utilised to design the integrated MPC controller.

I have some issues to discuss with the authors:

  • It is better to review more recent studies about the integrated yaw and roll stability control, particularly the MPC control methods.
  • Please cite some references to show how to estimate the longitudinal velocity, lateral velocity and vertical velocity. 
  • Is the proposed algorithm computationally efficient?
  • For implementation, what about the robustness issues against model uncertainties of different road frictions or vehicle parameter variation? Also the measurement noises and estimation uncertainties?
  • For Equation (39), the rotational inertia of the wheel as shown in Equation (18) is not considered. Is it a valid assumption?
  • Please clarify how to determine the threshold values of load transfer ratio?

Reviewer 4 Report

This paper addresses the problem of controlling a vehicle in the event of a sudden tire blowout. The vehicle is modeled as a 14 degrees of freedom mathematical representation and a control strategy with two control levels. The upper supervisory level determines the best suitability control mode in a lookup table approach that contains some relevant scenarios for the vehicle. The lower control level contains the predictive controller which minimizes a cost function based on the future estimates of the error and inputs. 

In general, the topic is interesting and the paper needs revision to improve its readability. 

* Particularly it is important to make clear which is the contribution of the paper. Is it the 14-dof model or the control strategy?

* Please include more relevant recent (2019-21) references in the literature analysis in order to better frame your contribution. 

* The 14-dof model is this the first time is presented? 

*The model is validated using experimental data from [7], however, the paper does not explain the parameterization of the models 14-dof and 8-dof in order to be the same as the one presented in [7].

*Define all the variables and acronyms on first use, e.g. C.G. Check the paper thoroughly. 

*Avoid the use of double letters to name a single variable, e.g. fl for front-left; LTR for load transfer ratio.  

*The model represents only the car dynamics after the tire blow-out? Why not consider a complete model of the car and treat the blow-out as a fault in order to use fault diagnosis and fault-tolerant control strategies? e.g. A review of convex approaches for control, observation, and safety of linear parameter varying and Takagi-Sugeno systems, Processes, 2019, 7(11), 814. Please discuss. 

*Can the model represent the dynamics of the car irrespective of where the blown tire is located?

*The discretization of the model is one of the most basic (Euler method), why not use a more accurate discretization?

*The computational load of the predictive control strategy is in the dof of the online optimization rather than how the state is obtained (estimated or sensed). Then the paragraph on lines 426-433 is not true.

*The supervisory scheme based on a lookup table can be improved by using a multi-objective control strategy? 

* The predictive control strategy is not correctly described. Please state the control law explicitly. How the minimization problem subject to constraints (47) is converted to an optimization in terms of the decision variables.

* Why the optimization problem (47) does not include weights in its formulation?

* Include the stability analysis of the overall control loop, including with a healthy tire and the scenario with a blown-up tire. 

* In the simulations, the sampling time is 0.005s and the prediction horizon is 5, i.e. the prediction horizon is 0.025 s. How does it relate to the time constant of your model? does the prediction horizon cover the time constant or how it was tuned? 

* What about the control horizon? The number of future input changes calculated. These are the dof of the optimization problem and the ones determining the online computational complexity of the algorithm.  

In summary, the problem is interesting, but the paper needs to be improved. The contribution needs to be clarified and the description of the predictive control strategy needs to be considerably improved: include details in how the control law is obtained and how it is applied to the vehicle. Include the stability analysis of the overall control loop.  The language also needs to be polished, the manuscript has a number of typos and grammatical issues.

Round 2

Reviewer 4 Report

I am happy with the revised paper. I have no more comments.

This manuscript is a resubmission of an earlier submission. The following is a list of the peer review reports and author responses from that submission.

Round 1

Reviewer 1 Report

The paper deals with the stability control of road vehicles in case of tire blow-out is presented. A novel controller is presented and its effectiveness is assessed through simulations. The following comments should be addressed.

  1. In the Introduction (lines 65-67) it is stated that the EDVSM model has 15 DOFs. However, it is also stated that it has 6+4+1 DOFS. Please, clarify.
  2. Equations 1-5. Apparently, not all the quantities are explicitly defined (even if their meaning can be derived from the schematics and from the nomenclature list at the end of the manuscript). For instance, what does the subscript j stand for? Please, declare explicitly all the parameters.
  3. Line 177. Reasonably, Cs is the longitudinal stiffness. In addition, should subscript i also added?
  4. Line 183. It may be advisable declaring the quantity x_t0 in a separate equation, not within the paragraph.
  5. Section 4.2, lines 386-402, and Figure 8. The description of the controller logic should be improved. In particular, it is not clear if the yaw, roll and cruise control modes can be all active at the same time. The diagram should be updated accordingly.
  6. Why is the EDVSM model only used in Section 3 and not to test the proposed controller in Section 5? Please, provide some comments on this aspect in Section 5.
  7. Might the proposed controller be also adapted to electric industrial vehicles, such as forklift trucks, which may also suffer stability issues (e.g. see Refs [a,b], listed below)? Some considerations on this aspect may be included, and the mentioned works may be added to the References as examples.

[a] Martini, A.; Bonelli, G.P.; Rivola, A. Virtual Testing of Counterbalance Forklift Trucks: Implementation and Experimental Validation of a Numerical Multibody Model, Machines 2020, 8(2):26. doi: 10.3390/machines8020026

[b] Rebelle, J.; Mistrot, P.; Poirot, R. Development and validation of a numerical model for predicting forklift truck tip-over, Vehicle System Dynamics 2009, 47(7), pp. 771-804. doi: 10.1080/00423110802381216

Author Response

Response to Reviewer 1 Comments

Point 1: In the Introduction (lines 65-67) it is stated that the EDVSM model has 15 DOFs. However, it is also stated that it has 6+4+1 DOFS. Please, clarify.

Response 1: Thank you for your comment. The 15 DOF EDVSM model includes 6 DOF for the vehicle body, 4 DOF for the suspension system, 4 DOF for the wheel rotation and 1 DOF for the steering wheel. This statement has been includes in the introduction part.

Point 2: Equations 1-5. Apparently, not all the quantities are explicitly defined (even if their meaning can be derived from the schematics and from the nomenclature list at the end of the manuscript). For instance, what does the subscript j stand for? Please, declare explicitly all the parameters.

Response 2: Thank you for your comment. All the quantities are clearly defined in equation (1)-(29) in the revised paper. Furthermore,  should be rewritten as .

Point 3: Line 177. Reasonably, Cs is the longitudinal stiffness. In addition, should subscript i also added?

Response 3: Thank you for your comment. In the revised paper,  has been rewritten as .

Point 4: Line 183. It may be advisable declaring the quantity x_t0 in a separate equation, not within the paragraph.

Response 4: Thank you for your comment. In the revised paper,  and  have been rewritten as a separate equations.

Point 5: Section 4.2, lines 386-402, and Figure 8. The description of the controller logic should be improved. In particular, it is not clear if the yaw, roll and cruise control modes can be all active at the same time. The diagram should be updated accordingly.

Response 5: Thank you for your comment. In the revised paper, the description of controller logic in section 4.2 and Figure 8 have been revised to better present the logic of different modes selection, especially it is clear shown that the yaw stability, roll stability and cruise control modes can be all activated at the same time.   

Point 6: Why is the EDVSM model only used in Section 3 and not to test the proposed controller in Section 5? Please, provide some comments on this aspect in Section 5.

Response 6: Thank you for your comment and we understand your concern. The EDVSM model is suggested in reference [5]. In reference [5], the simulation results of EDVSM model during tyre blowout without controller applied are compared with the actual experimental results.  Unfortunately, there is no clear description of the EDVSM model equations so we can not built EDVSM model by ourselves. We can only validate our proposed 14 DOF model against the EDVSM model simulation result in the same simulation manoeuvre of tyre blowout in [5] in section 3, but we cannot built the EDVSM model and generate the simulation results after MPC controller implemented in section 5.

Point 7: Might the proposed controller be also adapted to electric industrial vehicles, such as forklift trucks, which may also suffer stability issues (e.g. see Refs [a,b], listed below)? Some considerations on this aspect may be included, and the mentioned works may be added to the References as examples.

[a] Martini, A.; Bonelli, G.P.; Rivola, A. Virtual Testing of Counterbalance Forklift Trucks: Implementation and Experimental Validation of a Numerical Multibody Model, Machines 2020, 8(2):26. doi: 10.3390/machines8020026

[b] Rebelle, J.; Mistrot, P.; Poirot, R. Development and validation of a numerical model for predicting forklift truck tip-over, Vehicle System Dynamics 2009, 47(7), pp. 771-804. doi: 10.1080/00423110802381216

Response 7: Thank you for your suggestions. The stability issues of the forklift trucks during tyre blow-out have been added in the first paragraph in the introduction part and the above references have been added in the revised paper.

Reviewer 2 Report

This manuscript has proposed an interesting topic about integrated coordinate control based on 14 DOF MPC during tyre blowout conditions. The simulation performance of the 14 DOF vehicle model during tyre blowout has been validated with the EDVSM model, which has been applied in the internal model of MPC. The whole structure of the manuscript is clear and the manuscript is well written. However, I still have some minor issues:

(1) How long is the prediction horizon for MPC? What is the sampling time?

(2) Please redraw Figure 8 to show that the yaw plane stability control mode is disabled when the critical roll stability control mode is selected.

(3) Control efforts should be presented for three sets of simulations.

(4) Please indicate what the required sensor measurements are for implementation?

(5) Please explain why full brake is selected for critical roll stability mode. Is there any reference to support this decision?

(6) The Literature Review should be clarified sufficiently. There are some related paper should be referenced, for example

[1] Yin, C., et al. "Nonlinear Model Predictive Control for Path Tracking Using Discrete Previewed Points." 2020 IEEE 23rd International Conference on Intelligent Transportation Systems (ITSC) IEEE, 2020.

[2] Chen, L. , et al. "Lateral control using LQR for intelligent vehicles based on the optimal front-tire lateral force." Qinghua Daxue Xuebao/Journal of Tsinghua University (2021).

Reviewer 3 Report

The abstract states: ... a comprehensive 14 degrees-of-freedom (DOF) vehicle dynamics model is first proposed to describe the vehicle yaw plane and roll plane dynamics performance after tyre blow-out.

However, the model presented here incorporates crude approximations which are not discussed and much worse not even reported. It is based on erroneous equations of motion which represent a mixture of a quasi-static approach combined with some pseudo three-dimensional dynamics roughly composed of independent rotations about three axis.

According to Section 2.4 the tyre blow-out is modeled not by the tyre itself but by a sudden increase of the rolling resistance which induces an additional yaw moment. This approach is rather based on hearsay than on physics. A blow-out significantly changes the tyre properties. A fast but not sudden decrease of the vertical stiffness results in an increased rolling resistance and an decreasing flank stiffness. A sufficiently accurate tyre model (most likely not the Dugoff model applied here) should be able to approximate all tyre forces and torques applied in the contact patch by changing the tyre model parameters accordingly. No additional yaw moment applied like a deus ex machina is required then.

The manuscript refers to the three-dimensional Engineering Dynamics Vehicle Simulation Model (EDVSM) which, compared to the model presented here, just adds an additional DOF for the steering wheel. The simulation results of Figure 2 are supposed to represent a tyre blow-out where no steer input is involved. Hence, the results of the 14 DOF model and the EDVSM model (15 DOF model) should practically be identical. Therfore the keen and non-scientific statement in Section 3 "Although there are some mismatches compared with EDVSM model, the yaw rate and lateral acceleration responses of 14 DOF model still shows similar response as EDVSM model" should be changed to "The mismatches compared with EDVSM model are alarming and require a careful and detailed review of the vehicle model and probably of the control strategy in addition". 

Although the title refers to "Combined Yaw and Roll Stability", Figure 2 does not provide a comparison of the 14 and 15 DOF models with respect to the roll motions. Most likely, because there is no similarity at all.

Some remarks to the vehicle model proposed here:

In a classical approach, c.f. the paper "State-of-the-art and challenges of railway and road vehicle dynamics with multibody dynamics approaches" https://doi.org/10.1007/s11044-020-09735-z which is not cited here, a three-dimensional passenger car model with 14 DOF consists of the chassis or sprung mass with 6 DOF, 4 knuckles or unsprung masses with 4*1=4 DOF, and 4 wheels with 4*1=4 DOF which according to the principles of mechanics results in a set of nonlinear and coupled differential equations.

A vehicle dynamics model put together by simplified yaw plane and roll plane motions, as done here, presents a rather crude and more or less quasi-static approximation to a fully nonlinear and dynamic motion of a vehicle. Thoroughly simplified models may be used for basic studies, as performed nearly 50 years ago in: Mitschke, M.: Dynamk der Kraftfahrzeuge. Springer, Berlin (1972). https://doi.org/10.1007/978-3-662-11585-5, but they definitely are not suitable to investigate the highly dynamics and nonlinear response of a vehicle to tyre blow-out.

The equations of motion presented in Equations (1a) - (1f), are driven by the forces Fxsij, Fysij, Fzsij, and Fdzij. The latter are specified as "load transfer forces" in (7a) and (7b). But its sum, which should be applied to the sprung mass in (1c), consisting of Fdz = Fdzfr + Fdzfl + Fdzrr + Fdzrl is vanishing according to (7a) and (7b). So, why is Fdzij included in (3a) at all?.

Most likely Fxgsi, Fygsi introduced in (3a) and (3b) correspond with Fxsij, Fysij used in (1a) and (1b). But then, the equations of motion (1a) to (1f)
are driven by the tyre forces, which means that the dynamics of the unsprung masses (wheels+knuckles) are neglected here. But then, Equations (1a) - (1f) are just a very crude approximation to the dynamics of the overall vehicle which consist of one sprung and four unsprung masses. 

Equations (16) and (19) provide simple models for the wheel dynamics and the vertical motions of the unsprung mass. The latter neglects the dynamic coupling to the motions of the sprung mass.

Equations (1d) to (1f) are supposed to model the rotational dynamics of the sprung mass. They erroneously consist of "torque balances" about the roll axis and the body-fixed y-, and z-axis supplemented by some pseudo-inertia torques.  However, as illustred c.f. at https://en.wikipedia.org/wiki/Euler%27s_equations_(rigid_body_dynamics) the three-dimensional rotation of a rigid body also contains gyroscopic torques, which may only be neglected at small angular velocities. For sure, the heavy impact of a tyre blow-out will cause significant roll, pitch and yaw rates simultaneously.

Besides that, the roll axis defined by hf and hr and the y- and the z-axis define no valid body-fixed reference system, because the do not intersect in one point and are not mutually perpendicular if hf differs from hr as common to most vehicles.

It seems that Equations (1d) to (1f) and (19) do not contain the impact of ant-roll bars which are a standard equipment of vehicles and have a significant influence on the vehicle stability.

The manuscript uses Dugoff tyre model. The model is 50 years old, c.f.
Dugoff, H., Fancher, P., Segel, L.: An Analysis of Tire Properties and Their Influence on Vehicle Dynamic Performance. SAE Technical Paper 700377 (1970). Although it has been updated in https://www.researchgate.net/publication/274565927_A_Dynamic_Model_for_TireRoad_Friction_Estimation_under_Combined_LongitudinalLateral_Slip_Situation. it is still limited to the quasi-static tire behavior.
There is no evidence that it is capable to model the highly dynamic response of a tyre blow-out. A variety of tyre models of different compexities, ranging from handling tyre models with a first order dynamics to structural or FE-model, can be found in
https://iavsd.org/wordpress/wp-content/uploads/Proceedings-of-the-4th-International-Tyre-Colloquium_Gruber_Sharp_2015.pdf

To summarize:

The presented vehicle model is poor and erroneous. As a consequence, the Control Framework for Combined Yaw and Roll Stability proposed here is questionable and probably just operating with the erroneous equations but of no use in reliable vehicle models or in practice.

Round 2

Reviewer 3 Report

The revised paper just contains minor changes, although a major revision was requested.

Major problems are postponed to the future by phrases like

... it will be conducted as our further study in the near future 

... which required further investigation

...  we will consider more complex vehicle dynamics model in the future 

... In the future, we will discuss the performance based on the different tyre models.

Hence, the revised manuscript is still based on a crude model approach which is not validated properly and is definitely not state-of-the-art in vehicle dynamics.